# Altered differentiation of endometrial mesenchymal stromal fibroblasts is associated with endometriosis susceptibility

Brett D. McKinnon [1,5✉], Samuel W. Lukowski [2,5], Sally Mortlock[2], Joanna Crawford[2], Sharat Atluri[2], Sugarniya Subramaniam[2], Rebecca L. Johnston [2,3], Konstantinos Nirgianakis[1], Keisuke Tanaka[4], Akwasi Amoako[4], Michael D. Mueller[1] & Grant W. Montgomery [2]

Cellular development is tightly regulated as mature cells with aberrant functions may initiate pathogenic processes. The endometrium is a highly regenerative tissue, shedding and regenerating each month. Endometrial stromal fibroblasts are regenerated each cycle from mesenchymal stem cells and play a pivotal role in endometriosis, a disease characterised by endometrial cells that grow outside the uterus. Why the cells of some women are more capable of developing into endometriosis lesions is not clear. Using isolated, purified and cultured endometrial cells of mesenchymal origin from 19 women with ($n = 10$) and without ($n = 9$) endometriosis we analysed the transcriptome of 33,758 individual cells and compared these to clinical characteristics and in vitro growth profiles. We show purified mesenchymal cell cultures include a mix of mesenchymal stem cells and two endometrial stromal fibroblast subtypes with distinct transcriptomic signatures indicative of varied progression through the differentiation processes. The fibroblast subgroup characterised by incomplete differentiation was predominantly (81%) derived from women with endometriosis and exhibited an altered in vitro growth profile. These results uncover an inherent difference in endometrial cells of women with endometriosis and highlight the relevance of cellular differentiation and its potential to contribute to disease susceptibility.

[1] Department of Obstetrics and Gynaecology, University Hospital of Berne, Berne, Switzerland. [2] The Institute for Molecular Bioscience, The University of Queensland, Brisbane, Australia. [3] Department of Genetics and Computational Biology, QIMR Berghofer Medical Research Institute, Brisbane, Australia. [4] Department of Gynaecology, Royal Brisbane and Women's Hospital, Brisbane, Australia. [5]These authors contributed equally: Brett D. McKinnon, Samuel W. Lukowski. ✉email: brett.mckinnon@dbmr.unibe.ch

Every tissue is a complex biological system consisting of heterogenous cell mixtures that developed through cellular differentiation and maturation. Tight regulation of these processes are required to maintain homeostasis. They can be influenced by both cell-autonomous and non-autonomous factors. Differentiation is contingent on stochastic interactions and subject to biological variability[1]. Any alterations in the differentiation or maturation process may give rise to subtle biological variations, introduce heterogeneity that leads to functional consequences and influence disease susceptibility.

The endometrium is the reproductive tissue that lines the uterus and plays a critical role in reproduction. It is unique in that it is consistently shed and regrown each month, generating up to 10 mm of new mucosa. Over the reproductive life of a woman, it undergoes >400 cycles of growth, differentiation and shedding[2]. The endometrium is made up of luminal and glandular epithelial cells supported by a vascularised stroma with immune infiltration. Endometrial stromal fibroblasts are regenerated from endometrial mesenchymal stem cells (eMSC) located at perivascular locations in basalis, not shed during menstruation[3]. In the endometrium, local niche effects largely restrict eMSC differentiation into mesodermal stromal fibroblasts[4] that subsequently differentiate into secretory decidual cells under hormonal stimulation. As a highly regenerative tissue, the endometrium has the potential each month for aberrant differentiation to occur.

Endometriosis is a reproductive disorder characterised by the growth of endometrial tissue outside the uterus. Endometrial cells are thought to enter the peritoneal cavity through retrograde menstruation[5]. Up to 80% of women experience retrograde menstruation, however, only a proportion develop endometriosis[6]. Inherent factors in the cells of some women must underlie an increased disease susceptibility. A number of studies have reported differences between the endometrium of women with and without endometriosis, although differentially expressed genes have been difficult to consistently replicate[7], potentially due to the dynamic nature of the tissue. Recent large scale genome-wide gene expression studies on endometrial tissue also reported no significant difference in gene expression between women with and without endometriosis once the menstrual stage and multiple testing correction was applied[8,9]. Although increasing in power, these studies are limited by the complex milieu of cells and cellular states that may mask subtle differences.

Cells of mesenchymal lineage are some of the most abundant in the endometrium and are strongly implicated in endometriosis pathogenesis. SUSD2+ eMSC have been identified in both peritoneal and menstrual fluid and may have a key role in the establishment and proliferation of ectopic endometrial tissue[10] through their clonogenic and multipotent differentiation capacity[5]. eMSC from women with endometriosis had impaired in vitro decidualisation[11], as well as altered activation of signalling pathways during decidualisation[12,13].

To identify inherent variation in the endometrium that could underlie endometrial susceptibility we therefore assessed gene expression of individual mesenchymal-derived cells isolated and cultured from the endometrium and their association with clinical parameters and in vitro growth. We identified eMSCs and two distinct endometrial stromal fibroblasts populations generated by divergent differentiation from MSC to their mature cell state, one of which was characterised by gene expression profile indicative of an altered immune state and was found significantly more frequently to have been derived from women with endometriosis. This study links single-cell transcriptome data with both functional and clinical characteristics and uncovers a potential role for divergent mesenchymal-derived stromal fibroblast maturation to contribute to endometriosis susceptibility.

## Results

**Pure, single cells of mesenchymal origin were isolated from women with and without endometriosis.** Endometrial stromal cells isolated from endometrial biopsies were grown in culture and stored frozen (Fig. 1a). We selected 22 frozen samples for analysis and to ensure pure, viable cells of mesenchymal lineage from thawed preparations and conducted two-channel FACS sorting with forward and side scatter (Fig. 1b), propidium iodine (PI) exclusion (Fig. 1c) and platelet-derived growth factor β+ (PDGFB+) expression (Fig. 1d). The mean cell concentration after initial thaw was $2.19 \times 10^6$; (range $0.172 \times 10^6$–$3.46 \times 10^6$) with 90.0% remaining viable after the thawing process. One sample did not yield a sufficient cell concentration, and two samples had a final viability <80% and were not carried forward to single-cell analysis, resulting in a final 19 samples.

Of the resulting 19 samples, ten were derived from patients with endometriosis and nine were from women which had no endometriosis observed during surgery. For all 19 women (both cases and controls) no adenomyosis was observed during surgery. One control patient was diagnosed with subserosa uterine myomatosis and one case was diagnosed with intramural uterine myomatosis. Using the most severe form of the lesion to define endometriosis subtype[14], two endometriosis patients were classified as superficial peritoneal, two were classified as endometrioma (with one having an endometrioma only and the other an accompanying superficial lesion). The remaining six endometriosis patients were classified as DIE with four patients having a DIE lesion accompanied by a superficial lesion, one patient having a combination of a superficial peritoneal lesion, OMA and DIE lesion and one patient with an OMA and DIE lesion.

Serum progesterone measurements were available for all 19 patients and age and BMI were available for 18 patients. Nine endometrial stromal cell preparations were isolated during the proliferative phase (7× cases and 2× controls), four were isolated during the periovulatory period (1× case and 3× controls) and six during the secretory stage (2× cases and 4× controls). No significant difference in menstrual cycle phases was observed between cases and controls (Supplementary Table 1). The average age and BMI of all patients were $35.17 \pm 1.71$ and $24.16 \pm 1.06$ respectively. There was no significant difference in age (case = 35.17 vs control = 35.0, $p = 0.939$), although there was a significant difference in BMI (case = 24.16 vs control = 27.29). Cells were frozen down between passages 4 and 7, with p4 = 4 (2× case, 2× control), p5 = 4 (0× case, 4× control), p6 (7× case, 1× control) and p7 = 1 (1× case, 0× control). A chi squared comparison indicated a significant difference between cases and controls ($p = 0.026$) with a slightly higher mean passage number for stromal cells from endometriosis cases, reflective of an enhanced growth profile. Through this process, we were able to prepare single endometrial cells of mesenchymal origin from 19 patients that were subsequently analysed through transcriptome-wide gene expression profiling.

**Single-cell RNA-sequencing and assessment of cluster resolution identified three consistent subtypes of endometrial mesenchymal cells.** In order to increase the size of the patient cohort, we used a multiplexing approach to combine samples from different patients into four microfluidic runs of the 10X Genomics Chromium Platform (P1: Patients 1–6, P2: 7–11, P3: 12–16 and P4: 16–19) (Fig. 1a). We obtained sequencing data from the four scRNA-seq libraries constructed from our four pools of 19 endometrial stromal cells (ESC). Endometriosis cases and controls were distributed across each pool (P1: Control = 2,

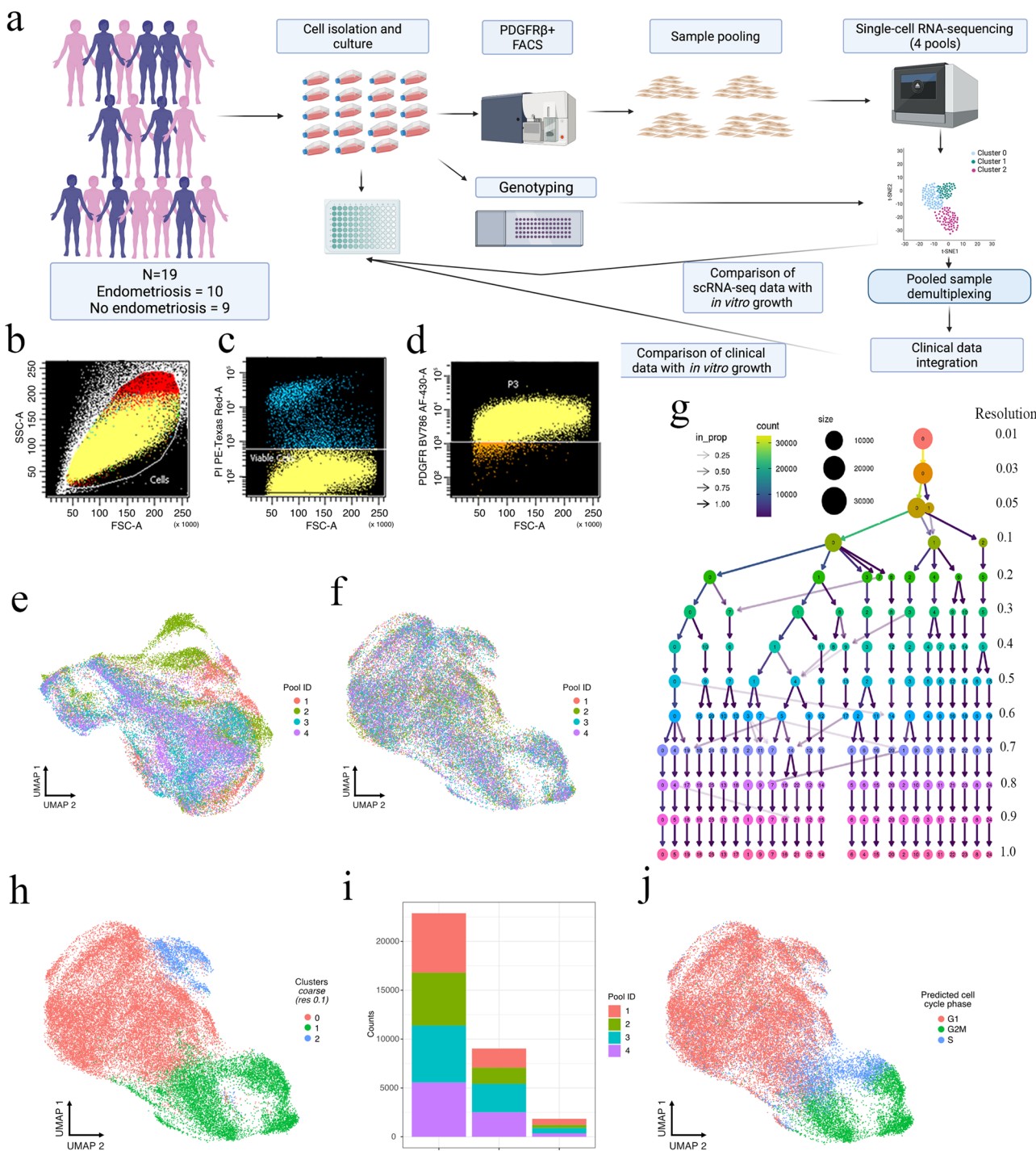

**Fig. 1 Experimental design and quality control for high throughput single-cell RNA sequencing of purified endometrial mesenchymal cells.**
**a** Endometrial biopsies were isolated, from 19 different women and cultured. In vitro growth assays, DNA isolation for genotyping and PDGFRβ + FACS purification was performed. Purified cells from nineteen samples were pooled into four lanes and run on the 10X Chromium controller and scRNA-seq data was analysed and cell clustering was performed. Samples were assigned to their source individual and clusters compared to clinical data. Both scRNA-seq data and clinical data were compared to in vitro growth. Mesenchymal stromal cells were purified via FACS (**b**) forward and side scatter, **c** viability and **d** PDGFRβ + expression. UMAP plot distribution of scRNA-seq data was determined both **e** pre- and **f** post harmony correction of between-pool variations introduced through technical variations. **g** Clustree analysis was used to determine the most stable level for cellular clustering. **h** UMAP plot of integrated scRNA-seq data at a clustering resolution of 0.1 identified three distinct clusters with minimal subsequent mixing at a finer resolution. **i** At a cluster resolution of 0.1, cell numbers from each pool in each cluster remained stable suggesting clusters were formed from biological differences rather than technical effects. **j** UMAP plot of scRNA-seq data labelled by cell cycle phase inferred by the CellCycleScoring function in Seurat. **a** Created with BioRender.com.

Case = 4, P2: Control = 3, Case = 2, P3: Control = 3, Case = 2, P4: Control = 2, Case = 2).

For each pool, the number of cells obtained were 10,411 (P1), 9869 (P2), 10,894 (P3) and 9808 (P4) making a total of 40,982 cells. Demuxlet identified 3982 doublets that were randomly distributed across each pool (Supplementary Fig. 1a–d). We also excluded 358 ambient cells, 2884 with >10% mitochondrial DNA that was considered stressed or dying cells, and cells with either very high (>6500), or very low (<200) numbers of expressed genes. From the initial 40,982 cell dataset we retained 33,758 high quality singlets for analysis with an average read depth of 58,541 (P1), 58,277 (P2), 53,825 (P3) and 55,836 (P4) for each pool. We detected a median of 19,803 (P1), 20,095 (P2), 20,651 (P3) and 22,472 (P4) unique molecular identifiers (UMI) per cell with the total number of genes with measurable expression of 21,885 (P1), 21,736 (P2), 22,178 (P3) and 21,996 (P4). In total, 20,590 unique genes were identified across all four pools with the median number of genes expressed per cell as 3780 (P1), 3761 (P2), 3941 (P3) and 4050 (P4). After doublet-filtering and quality control, between-pool batch effects were corrected using Harmony (Fig. 1e, f).

We next investigated transcriptome similarity and assessed whether potential cellular subtypes were present through unsupervised Louvain clustering using Seurat v3.0.2[15]. A critical step in deriving relevant data from single-cell datasets is selecting the appropriate resolution for clustering. Increasing resolution increases cell clusters, although potentially at the expense of biological relevance (Supplementary Fig. 2). Using the clustree[16] package, we produced a cluster tree with 13 levels of resolutions ranging from 0.01 to 1.0 (Fig. 1g) to visualise the similarity between cells at multiple resolutions and track how cells move between clusters as the resolution is varied. This package uses a hard clustering algorithm to cluster data at multiple resolutions producing a set of cluster nodes, the overlap between clusters is used to build edges and the resulting graph represents how each cluster relate to each other, which are distinct and which are unstable. This allows the visualisation and exploration of all possible choices[16]. At a coarse resolution (0.1), three distinct nodes were identified that established stable clusters with minimal movement across nodes at increasingly finer resolutions. At this coarse resolution, there was also a distinct spatial separation for each cluster (Fig. 1h) and the number of cells from each experimental pool within each cluster was consistent (Fig. 1i). In contrast, clustering at a finer resolution (0.6) generated 20 clusters which lacked distinct spatial resolution and revealed mixing between clusters and possible over clustering resulting from technical artifacts (Supplementary Fig. 2). Through the inclusion of multiple patients, multiplexing of samples and deep sequencing of expressed genes we were able to sequence a dataset of sufficient size and quality to identify three stable clusters of distinct mesenchymal cells.

**Cell cycle scoring indicates cluster 1 harboured an increased portion of proliferating cells.** Genes can be periodically regulated during the cell cycle[17], influencing their transcriptome and affecting the ability to accurately cluster cells based on phenotype. To characterise the cell cycle for each cell, we calculated G1, G2M and S scores using the CellCycleScoring function in Seurat and human cell cycle phase gene expression profiles[18]. The majority of cells analysed (69.65%) showed a G1 phenotype, whilst 15.77% of cells were classified as G2M and 14.58% were classified as S phase (Fig. 1j). We observed an enrichment of the proliferating cells (G2M) in cluster 1 (54.94%). Cluster 1 also had an increased proportion of cells in S phase (31.94%) compared to clusters 0 (7.88%) and 2 (12.64%). The majority of cells in cluster 0 and 2

were classified as the quiescent G1 phase (90.80% and 84.11% respectively). This analysis indicated most cells analysed were in the G1 phase and the cell cycle was not directly associated with cell clusters.

**Differential gene expression between clusters reveals discrete signatures indicative of varied interaction with the microenvironment.** We next examined the differentially expressed genes (DEGs) that underlie these cluster differences (Fig. 2a). Setting a log fold change (logFC) > 0.25 and adjusted $p$ value < $1 \times 10^{-4}$ we found 152 significant DEGs between cells in cluster 0 and all other cells (Supplementary Data 1). A comparison between cluster 1 and all remaining cells found 707 DEGs (Supplementary Data 2), and cluster 2 (Supplementary Data 3) and all other cells found 113 DEGs. Spatial representation of three of the top DEGs in cluster 0 (IGFBP5; logFC = 0.984; adj. $p$ value < $1.0 \times 10^{-305}$; MMP11; logFC = 0.887; adj. $p$ value < $1.0 \times 10^{-305}$ and ACTA2; logFC = 0.728; adj. $p$ value < $1.0 \times 10^{-305}$) (Fig. 2b) revealed strong variation and non-synonymous distribution within the cluster, accompanied by low but consistent expression in the two other clusters. A similar spatial resolution was observed for UBE2S (logFC = 1.67; adj. $p$ value < $1.0 \times 10^{-305}$) with high expression in cluster 1 but low, consistent expression in the remaining clusters, although both PTTG1 (logFC = 1.833; adj. $p$ value < $1.0 \times 10^{-305}$) and UBE2C in particular (logFC = 2.02;; adj. $p$ value < $1.0 \times 10^{-305}$) showed a limited expression confined mostly to cluster 1. In cluster 2, MMP3 (logFC = 2.31; adj. $p$ value < $1.0 \times 10^{-305}$), CST1 (logFC = 2.19; adj. $p$ value < $1.0 \times 10^{-305}$ and MMP10; (logFC = 1.56; adj. $p$ value < $1.0 \times 10^{-305}$) showed significant differential expression, and both CST1 and MMP10 expression were limited mostly to cluster 2. We also performed differential expression analysis between specific cluster pairs (cluster 0 vs 1, 1 vs 2, and 0 vs 2) and detected 242, 246 and 17 significant DEGs, respectively (logFC > 0.5, adj. $p$ value < $1 \times 10^{-4}$; Supplementary Data 4, 5 and 6).

To gain further insight into the biological differences underlying the three clusters, we performed pathway analysis using the top 200 significant DEGs for each cluster using Reactome, KEGG and gene ontology databases (Fig. 2c). This revealed significantly enriched processes involved in extracellular matrix organisation (adj. $p$ value < $1.33 \times 10^{-12}$) and focal adhesion (adj. $p$ value = $1.1 \times 10^{-3}$) for cluster 0. For cluster 1 we observed significant enrichment of cell cycle (adj. $p$ value = $6.97 \times 10^{-12}$), progesterone-mediated oocyte maturation (adj. $p$ value = $4.28 \times 10^{-8}$) and oocyte meiosis (adj. $p$ value = $2.53 \times 10^{-7}$), whereas cluster 2 DEGs were enriched for extracellular matrix organisation (adj. $p$ value = $1.89 \times 10^{-14}$) but also antigen processing and presentation (adj. $p$ value = 0.0049), and allograft rejection (adj. $p$ value = 0.0389), indicating a potentially immune-reactive cell population (Fig. 2c). This analysis indicated gene expression profiles of cluster 0 and 2 were associated with the extracellular organisation, with cluster 0 focussed on adhesion, whereas cluster 2 may be influenced by the immune response. Cluster 1 appeared specifically related to reproductive development.

**Cell-type annotation confirmed gene expression signatures consistent with cells of mesenchymal origin.** Mesenchymal maturation can take multiple pathways leading to divergent progeny such as fibroblasts, adipocytes and smooth muscle cells[19]. To annotate our transcriptomically defined cell clusters we applied SingleR[20]. This utilises the transcriptomic signatures from the Human Primary Cell Atlas, a database curated from publicly available microarray datasets of human primary cells[21]. The analysis confirmed a close alignment with cells of mesenchymal lineage, albeit with variations in mesenchymal progeny distributed across the clusters (Fig. 3a). The five most prevalent cell

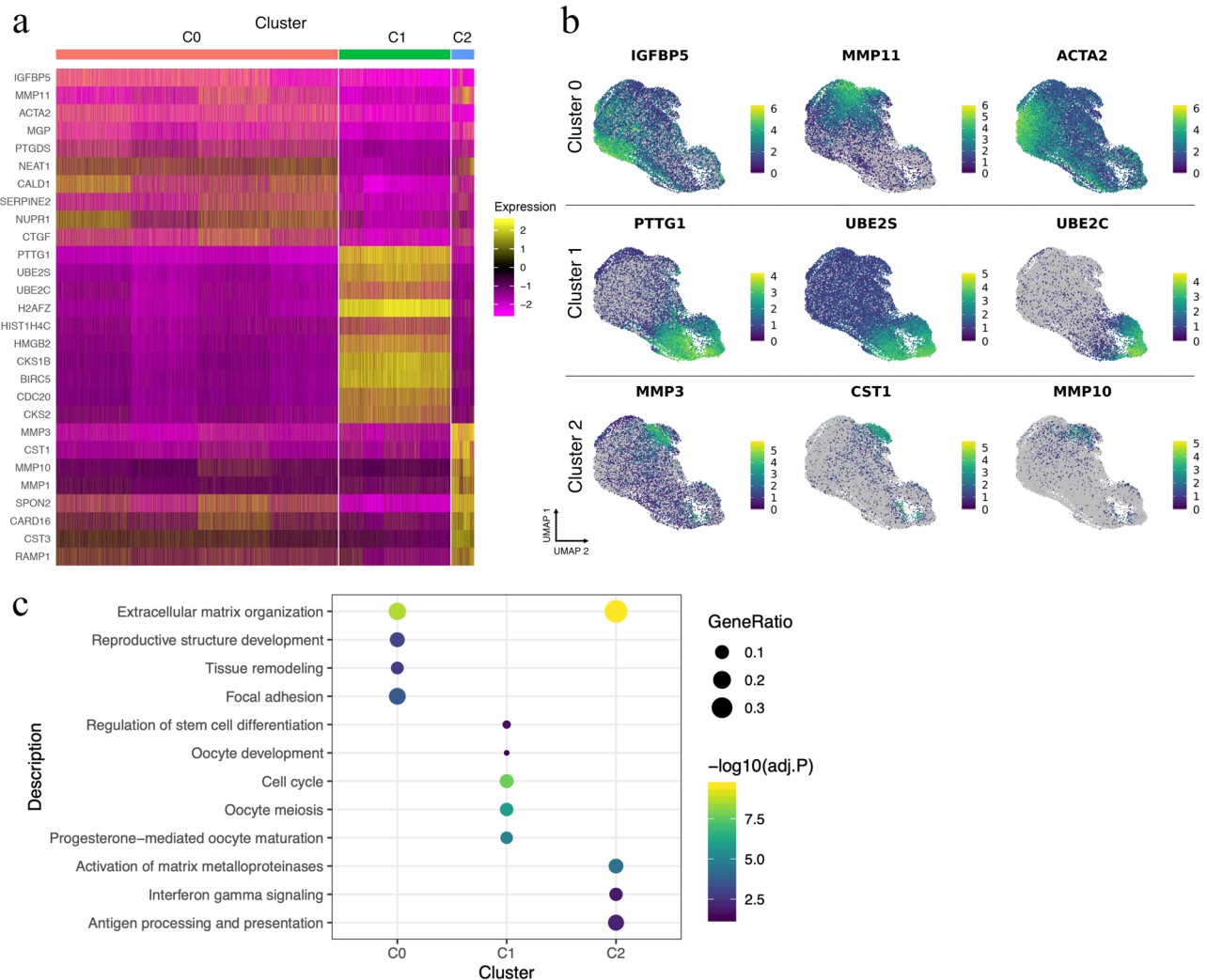

**Fig. 2 Differential gene expression and cell cycle characteristics of cell clusters. a** A heatmap representation of the differentially expressed genes (DEGs) across the different clusters. **b** UMAP plots coloured by log normalized expression of the top DEGs per cluster show strong gene expression in cells consistent with the location of the respective cluster, although for each gene the individual cellular expression is variable. IGFBP5, MMP11 and ACTA2 showed strong expression in cluster 0, but in distinct cells. Grey cells do not express the indicated gene. **c** Pathway analysis indicated a role for extracellular matrix organisation in cluster 0, oocyte maturation and meiosis and cluster 1 and activation of matrix metalloproteinases, interferon-gamma signalling and antigen processing and presentation in cluster 2.

types identified were fibroblasts (84.78%), mesenchymal stem cells (MSCs, 13.66%), smooth muscle cells (1.21%), induced pluripotent stem (IPS) cells (0.24%) and tissue stem cells (0.11%) (Fig. 3b).

Overlay of the different cell types with the clustering analysis revealed fibroblasts were the predominant cell type of cluster 0, representing (98.34%) of the cells in this cluster, with 0.93% identified as MSCs. As this was the largest cluster of fibroblast cells we designated this cluster 'fibroblast major'. Similarly, cluster 2 while distinct from cluster 0 was predominately fibroblasts (97.34%) with the inclusion of some MSCs (2.50%) and was subsequently termed 'fibroblast minor'. The clustering differences observed between the fibroblast clusters (clusters 0 and 2) could not be attributed to cell cycle differences (Fig. 2e). Cluster 1 was identified predominantly as MSCs (57.42%), and as such was named the 'MSC cluster', although 42.53% of cells within this cluster were also identified as fibroblasts (Fig. 3c). Analysis of gene expression signatures, therefore, supported the mesenchymal lineage of the cell dataset, but also indicated subtle differences exist within cells that can be used to delineate variations.

**Cell fate trajectory analysis identified the degree of mesenchymal differentiation for each cell**. To investigate dynamic biological processes within our dataset we applied pseudotime and RNA velocity analysis using Monocle 2 and scVelo[22–25]. This allowed the opportunity to study cellular differentiation or lineage progression by ordering individual cells along a trajectory, which can then be used to infer the state of individual cells in processes such as cell maturation (Fig. 3d). Overlay of the clustering data on the pseudotime trajectory predictions suggested a directional progression from cluster 1 (MSC cluster) as the root cell directing a cell fate lineage towards cluster 0 (fibroblast major) (Fig. 3e). This directional progression was also observed in the RNA velocity analysis (Fig. 3f), supporting the hypothesis that the cell differentiation pathway extends from cluster 1 (MSC cluster) to cluster 0 (fibroblast major). Cluster 2 (fibroblast minor) in contrast, was spread uniformly across the differentiation trajectory. Finally, we overlaid cell cycle information onto the trajectory plot and observed that most G2M phase cells, as well as the S phase cells corresponding to MSCs and the less differentiated fibroblasts consistent with the initial cell cycle analysis of each cluster and indicative of the higher proliferative ability of the MSCs (Fig. 3g).

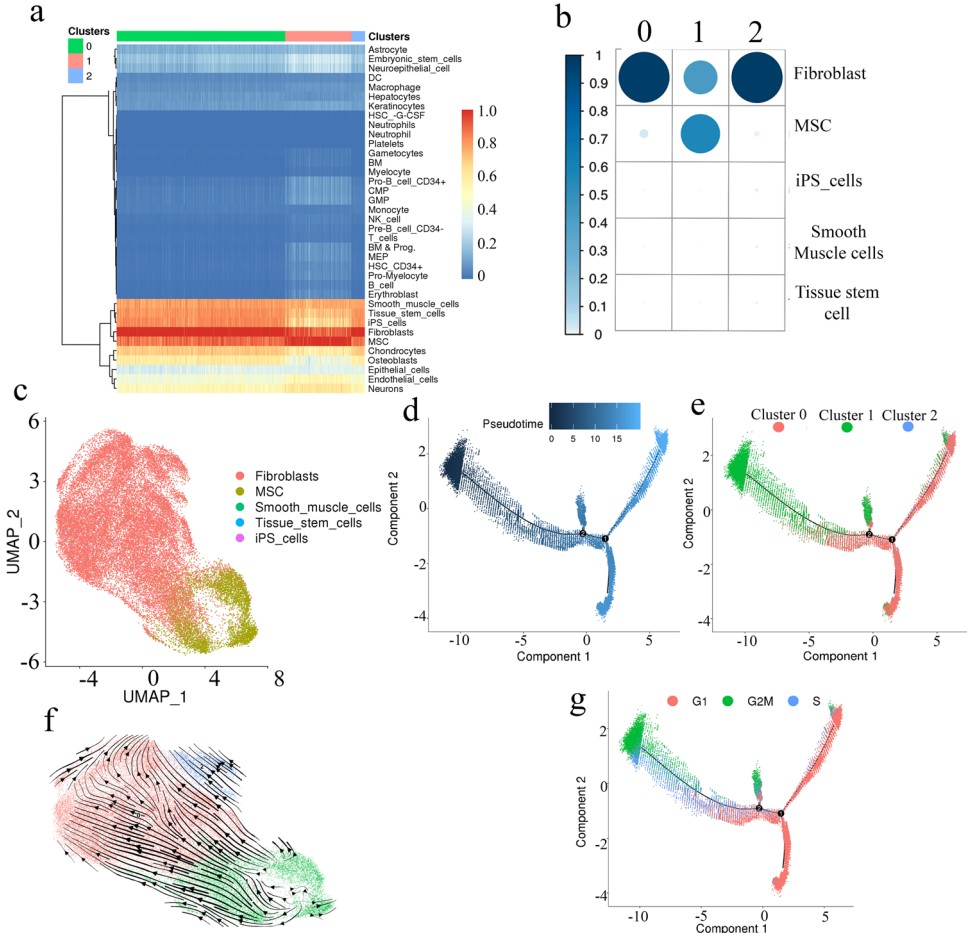

**Fig. 3 Cell type identification.** SingleR was used to assign cell types based on transcriptomic signatures. The strongest correlations were observed in red with the weakest in blue. **a** Heatmap visualisation of the transcriptomic signatures aligned strongly with cells of mesenchymal origin. This was consistent across all three clusters, although the actual identity varied across the clusters. **b** Distribution of the top cell type assignments across the three clusters. Circle size represents the proportion of each cell type identified in each cluster. Analysis indicated that fibroblasts were the predominant cells in cluster 0 and cluster 2. The majority of cluster 1 were MSCs, although 42.53% were considered fibroblast. **c** UMAP plot of scRNA-seq data labelled by cell type. Overlay of cell type on spatially resolved distribution depicts fibroblasts as the predominant cell in cluster 0 and cluster 2 and an association between MSCs with cluster 1. **d** Pseudotime cell fate trajectory analysis using Monocle 2 placed each cell on a continuum based on the similarity of the transcriptome. **e** An overlay of the clusters identified the majority of the cluster 1 MSCs as the root source directing cell fate lineage towards the fibroblast major cluster (cluster 0). The additional fibroblast minor cluster (cluster 2) was uniformly scattered across the continuum. **f** scVelo analysis supports the developmental trajectory direction of cluster 1 (green) to cluster 0 (red). A similar direction is also taken by cluster 2. **g** An overlay of the cell cycle analysis identifies the majority of G2M phase cells aligned with earlier trajectory and MSCs, transitioning to S phase followed by movement towards the majority G1 stage cells.

Comparison of the degree of gene expression change in each cell identified cluster 1 as the root cells extending to cluster 0 as the terminal differentiation. Cluster 1 existed across this trajectory and represented either incomplete or dedifferentiated mesenchymal cells.

**Deconvolution of pooled samples successfully assigned each cell to the patient of origin.** To ascertain whether cell types or cell clusters were associated with clinical phenotype we assigned each cell from the 33,758 cell dataset to the source patient. To demultiplex the individual patient samples in the four microfluidic pools we collected SNP genotyping information and used demuxlet[26] to assign each cell to the patient from which it was derived. Demuxlet uses statistical modelling to identify RNA-seq reads that overlap single nucleotide polymorphisms (SNPs). Using SNP data and imputation generated from genotyping the most likely donor for each cell can be identified. Demuxlet identified an average of 1229 SNPs (range = 13–3970) per cell

across pools; P1 (mean = 1264 SNPs/cell), P2 (mean = 1270 SNPs/cell), P3 (mean = 1213 SNPs/cell) and P4 (mean = 1176 SNPs/cell) (Supplementary Fig. 3a–d), allowing the confident assignment of 100% of the cells. Using this method the patient of origin for each cell and the number of cells analysed for each patient in each pool was identified (Fig. 4a).

**Comparison of gene expression profiles and cell clusters identified an association between the fibroblast minor cluster and endometriosis status in cultured stromal cells.** We wished to identify biological or clinical variables from our sample set that correlate with either cell type, or cell clustering. Splitting the cells based on the two major cell types identified, we assessed the correlation between gene expression and other variables including a number of passages, menstrual stage at the time of sample collection, patient age and endometriosis subtype. Almost all factors had a correlation close to zero and were well below the cut off value ($r^2 > 0.3$) to suggest any association with gene expression

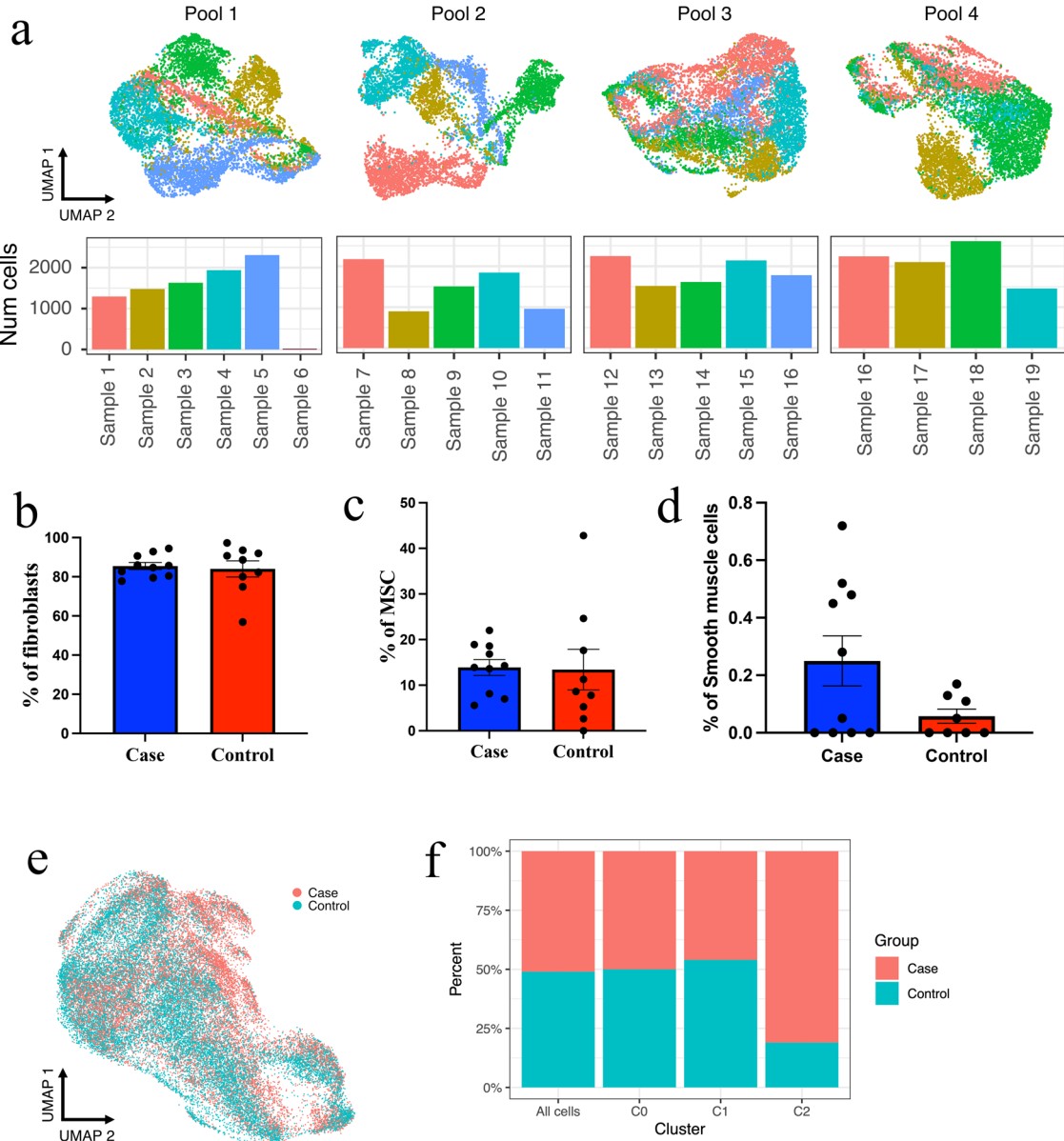

**Fig. 4 Cell type and cell cluster relationship to endometriosis status. a** Genotype data and demuxlet were used to deconvolute individual samples from the mixed pool of cells; 100% of cells were correctly assigned to the patient donor. In total 20 samples from 19 patients were loaded into 4 well of the 10X chromium controller. Overlaying clinical data and comparing the number of cells derived from women with endometriosis (cases, $n = 10$) and without endometriosis (controls, $n = 9$) using an unpaired t test found no significant difference between **b** the number of fibroblasts ($p = 0.7403$), **c** the number of MSCs ($p = 0.9185$) or **d** the number of smooth muscle cells ($p = 0.0723$). **e** UMAP plot of scRNA-seq data labelled by endometriosis status. **f** Percentage of cells assigned case or control status across all cells and per cluster (cluster 0, 1 and 2). A Chi-squared comparison between each cluster confirmed a significant association ($p < 0.0001$) between endometriosis cases and cluster 2. Error bars represent the standard error of the mean (SEM).

(Supplementary Table 2). We also assessed whether any of these characteristics could be associated with the cell numbers within each cluster. We found no significant association between the number of cells in each cluster with age (Cluster 0: $p = 0.3306$, Cluster 1: $p = 0.7670$, Cluster 2, $p = 0.8552$), or BMI (Cluster 0: $p = 0.4386$, Cluster 1: $p = 0.1352$, Cluster 2, $p = 0.9794$) nor was there an association between cell passage with any of the clusters identified (Cluster 0: $p = 0.1906$, Cluster 1: $p = 0.4724$, Cluster 2, $p = 0.1408$) (Supplementary Table 3).

We subsequently assessed whether the number of cells, including MSCs, fibroblasts and smooth muscle cells, were derived from either woman with or without endometriosis. In total, 16,650 cells (49.3%) were sourced from 9 women without endometriosis and 17,108 (50.7%) cells from 10 women with

endometriosis. The results indicate no significant variation in the percentage of fibroblasts from the endometrium of cases (84.93%) compared to controls (81.74%) (Fig. 4b). There was also no difference between the percentage of MSCs derived from the cases (14.48%) versus the controls (17.83%) (Fig. 4c). The percentage of smooth muscle cells derived from cases (0.24%) was larger compared to controls (0.08%) (Fig. 4d) although the difference did not reach significance ($p = 0.0738$).

Our data indicate that fibroblasts could be split into two distinct groups. We therefore also compared the number of fibroblast major (cluster 0), fibroblast minor (cluster 2) and MSC cluster (cluster 1) cells that were from women with and without endometriosis (Fig. 4e). The MSC cluster (cluster 1) had a total of 9033 cells of which 4199 (46.5%) were from women with

endometriosis and 4834 (53.5%) from women without endometriosis. The fibroblast major cluster (cluster 0) contained 22,881 cells with 11,418 (49.9%) from women with endometriosis and 11,463 (50.1%) from women without endometriosis. The fibroblast minor cluster (cluster 2) which contained 1844 cells consisted of 1491 (80.9%) cells from women with endometriosis and only 353 (19.1%) cells from women without endometriosis (Fig. 4f). A Chi-squared test indicated significantly more fibroblast minor cells (cluster 2) were from women with endometriosis ($p < 0.0001$). These data, therefore, indicate that the strongest association with gene expression profiles of the fibroblast minor cluster is the endometriosis status of the patient of origin.

**In vitro analysis confirmed cellular clusters displayed unique growth profiles.** Finally, we performed an analysis of cell growth rates for a continuous 100 h period using a subset of 11 cell preparations and the xCELLigence assay. We compared growth rates of individual cell preparations, endometriosis status, and the percentage of cell type (MSCs and fibroblasts) or cell subset (MSC cluster, fibroblast major, fibroblast minor). For individual preparations, the growth rates varied for both endometriosis cases and controls (Fig. 5a). Grouping cell preparation by endometriosis status showed variable rates of proliferation between cases and controls at different time points. Cells from controls had an initial (0–15 h) increased rate of proliferation with the growth rate eventually plateauing after 35 h. In contrast, the growth rate of cells from endometriosis cases continued to increase until the end of the incubation period (100 h) resulting in an increased number of cells from endometriosis cases, reflecting an increased capacity of continued growth although the difference was not significant ($p = 0.1737$) (Fig. 5b).

We also investigated the association between the contents of each cell preparation and growth rates by plotting the correlation between the percentage of each cell type, as determined by scRNA-seq and SinglR analysis and cell growth rates against time. The analysis revealed a positive association between MSC content and cell index that reached the strongest correlation between 8.25 and 9.75 h (Pearson's $r = 0.6364$, $p = 0.0402$) (Fig. 5c). Similarly, there was an opposite negative correlation with the percentage of fibroblasts and cell index between 8.5 and 9.5 h (Pearson's $r = -0.618$, $p = 0.0478$) (Fig. 5d).

As the scRNA-seq analysis identified two subsets of fibroblasts (fibroblast major and fibroblast minor) we further assessed the association between the per cent content of these cells in each cell preparation and growth rates. We found that each subtype displayed contrasting growth profiles, with the fibroblast major cluster (cluster 0) showing a non-significant positive association with growth rate and the fibroblast minor cluster (cluster 2) showing a significant negative correlation between 17.25 and 26 h (Pearson's $r = -0.681$, $p = 0.025$) (Fig. 5e). confirming the relative presence of each fibroblast influenced in vitro growth profiles at later time points after seeding compared to MSCs. This analysis indicated that variations in the in vitro cell cultures identified by gene expression signature can still influence cell growth patterns even after being placed in culture.

**MMP3 and ACTA2 are expressed by distinct sets of stromal cells in the endometrium.** To assess whether the gene expression markers of the fibroblast minor (MMP3) and fibroblast major (ACTA2) clusters could be detected at the protein level and whether these markers were expressed in distinct cells of mesenchymal origin in the endometrium, we probed endometrial samples from women with and without endometriosis, isolated at either the proliferative or secretory stage of the menstrual cycle

with specific antibodies for either MMP3, or ACTA2. Immunofluorescent images indicated only minimal MMP3 (red) expression in all endometrial samples (Supplementary Fig. 4). To exclude epithelial cell expression of these proteins we also performed co-staining with cytokeratin (green) a marker specific for epithelial cells. (Supplementary Fig. 4). Co-localisation with cytokeratin confirmed MMP3 expression in epithelial cells, however, some non-epithelial, stromal cells were also positive for MMP3. MMP3 expression appeared strongest in both endometriosis and control samples during the secretory stage with a slightly more prevalent expression in the endometriosis compared with control samples (Fig. 6a, b).

Incubation with serial sections of the same endometrial samples with ACTA2 (green) and cytokeratin (red) on adjacent slides (Supplementary Fig. 5) also confirmed the expression of ACTA2 in endometrial cells. Co-staining with cytokeratin (red) indicated the majority of ACTA2-positive cells were non-epithelial stromal cells, although in some cases epithelial cells also showed positive expression. The positive ACTA2 stromal cells appeared to be more common in the samples isolated from the proliferative phase compared to samples isolated from the secretory stage, irrespective of endometriosis status (Fig. 6c, d). No comprehensive quantification was performed to confirm these changes in expression. Importantly, using serial sections, we confirmed that both MMP3 and ACTA2 were expressed in endometrial stromal cells in the endometrium. Furthermore, we noted MMP3 expression appeared restricted to only a small number of endometrial stromal cells, consistent with the small cluster we observed at the gene level and that co-expression of MMP3 and ACTA2 in endometrial stromal cells was not detected when viewed under higher magnification (Fig. 6b and d). This supports the existence of independent endometrial stromal fibroblasts in endometrial samples.

## Discussion

Cellular heterogeneity both within tissue and within cell types is a key driver of tissue variation and disease susceptibility. To better understand endometrium and endometrial pathologies such as endometriosis, we assessed cell heterogeneity within the endometrial mesenchymal cell lineage and its association with clinical variables and in vitro cellular function. By profiling their gene expression at the single-cell level we identified three mesenchymal cell populations; a MSCs and two distinct stromal fibroblasts groups and charted their dynamic changes in gene expression. These data revealed the abundance of MSCs isolated was not related to endometriosis, but was associated with increased short-term in vitro growth. In addition, one fibroblast subpopulation displayed a gene expression profile indicative of dysregulated differentiation, and altered immune reactivity, and their percentage within each cell preparation inhibited in vitro growth rates. Importantly, this subpopulation was more likely to be derived from the endometrium of women with endometriosis compared to women without endometriosis. These results support a divergence in mesenchymal differentiation that alters fibroblast function and may predispose some women to endometriosis susceptibility.

Single-cell transcriptome analysis has previously revealed insights into endometrial cells, although it has not yet provided insight into clinical observations or endometrial pathologies. Previous analysis of the whole endometrium confirmed six distinct cell types; endothelial, epithelial (ciliated and unciliated), stromal and immune cells[27], although variations within cell types were not explored, potentially because it was not investigated and underpowered to do so with only 2149 cells and one biological replicate at each day of the menstrual cycle. In our study, we

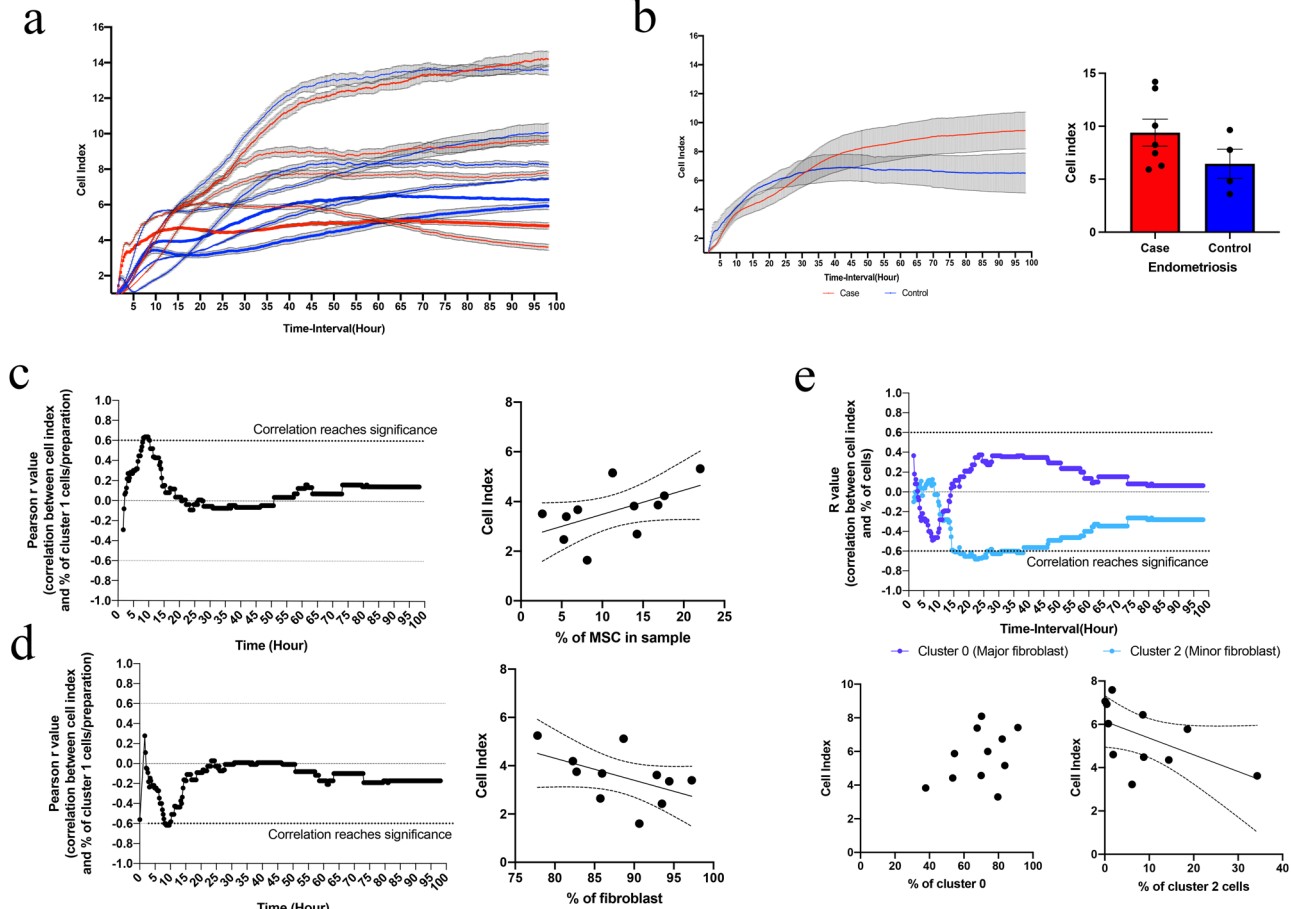

**Fig. 5 Relationship between cell types and cell clusters and in vitro behaviour.** We assessed cell growth rates over a 100 h period with the xCELLigence assay and compared the cell index to the percentage of cell types, or cell clusters identified in each cell preparation. **a** A comparison of the cell index for each individual preparation showed large variations in growth rates. **b** Categorisation of cell preparations based on endometriosis status showed varying rates of increase for both cases and controls across the 100-h growth assay. At the 100 h endpoint, the number of cells was higher for endometriosis cases ($n = 7$) compared to controls ($n = 4$), although not statistically significant when compared with an unpaired $t$ test ($p = 0.1737$). We subsequently compared the percentage of cell types in each preparation with the cell index using the Pearson correlation coefficient across all time points. This identified an increasingly positive correlation with cell index that reached significance (dotted line) with **c** MSC between 8.25 and 9.75 h after cell seeding ($P = 0.0402$), peaking at 9.25 (inset correlation graph). Conversely, for fibroblasts (**d**), a significant negative correlation ($p = 0.0478$) with cell index was observed at the same time point (inset correlation graph). **e** Lastly, a comparison with the fibroblast cell clusters revealed the association with cell growth was strongest between 17 and 26 h after initial seeding, with the slower growth rate of cluster 2 reaching significance (inset correlation graph) during this time period ($p = 0.0251$). Error bars represent the standard error of the mean (SEM).

focused on cultured endometrial cells selected via the mesenchymal marker PDGFRβ. Previous single-cell investigations of transcriptomic profiles of endometrial stromal cells showed that 64.9% of genes displayed consistent expression between both fresh and cultured cells[28]. While subtle variation mediated by niche environment will be lost during ex vivo processing, the use of primary cultured cells provides the opportunity to examine cell lineage differentiation in the absence of exogenous cues. It also provides the opportunity to perform experiments at scale, integrates clinical data and importantly assesses the transcriptomic relationship to in vitro growth characteristics.

Using these cultured samples we identified a significant proportion of MSCs remaining in all 19 culture preparations. MSCs reside in both the basalis and functionalis of the endometrium and are shed during menstruation potentially initiating endometriotic lesion growth[3]. In this dataset, we did not identify any differences in the MSC populations between women with and without endometriosis. Pseudotime trajectory and RNA velocity analysis indicated a variable differentiation from the MSC cluster to the fibroblast minor cluster that may have derived from

inherent variability within a subset of MSC cells, or a lack of appropriate niche signals in the culture environment. It has previously been shown transcriptomic variations in MSCs are inherited by daughter cells creating variation in the gene expression profile and biological function, potentially leading to increased disease susceptibility[11].

The fibroblast minor cluster was characterised by a transcriptome with potential for extracellular matrix organisation. Some of the most differentially regulated genes included matrix metalloproteinases (MMPs), *MMP3* and *MMP10*, both of which are within the stromelysin subclass of MMPs that have significant roles in extracellular remodelling of laminin fibronectin and gelatin (I–V) and collagens[29]. A genetic polymorphism in the promoter of *MMP3* is reported to be associated with endometriosis[30] and in the normal menstrual cycle, there is no *MMP3* expression in the proliferative phase, with an upregulation during the secretory stage that is significantly higher in women with endometriosis compared to women without[31]. Previous evidence reports the focal expression of *MMP3* in developing endometrium[32,33], data that would be consistent with our cluster

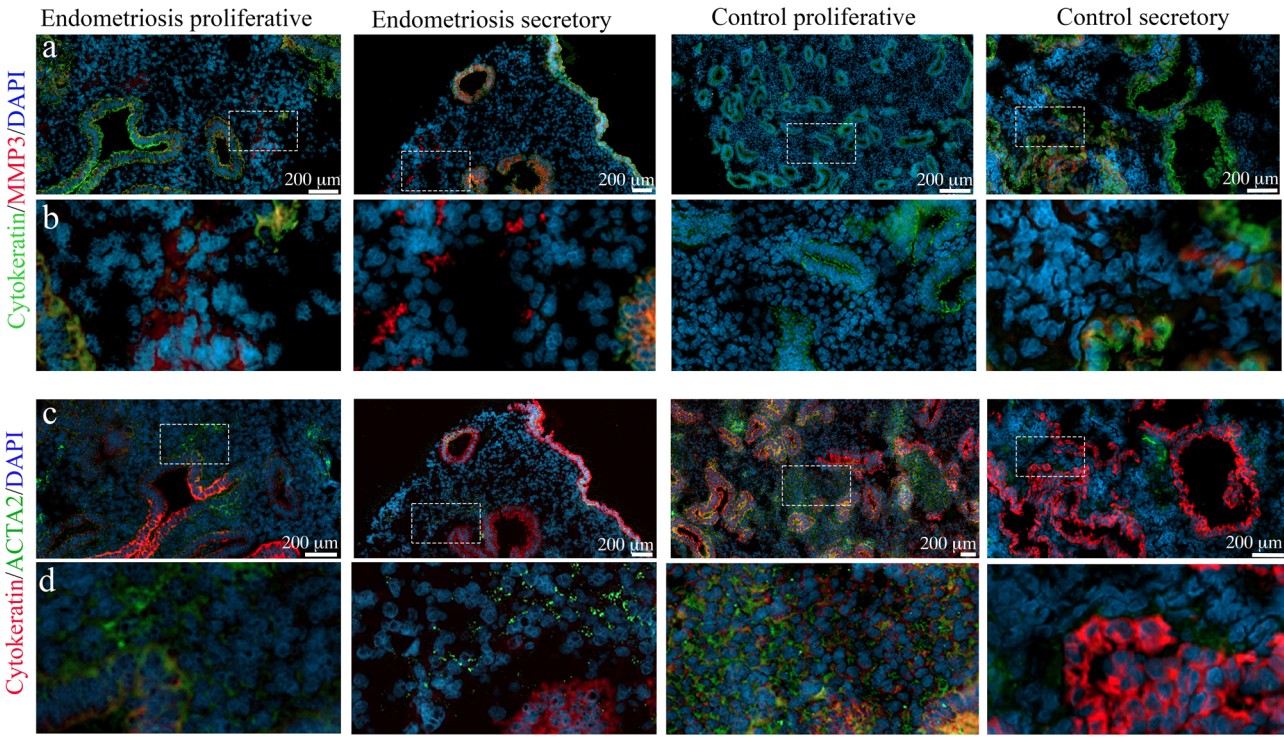

**Fig. 6 Protein expression of MMP3 and ACTA2 in the endometrium of women with and without endometriosis.** Using MMP3 (Upper panels) and ACTA2 (lower panels) specific antibodies and serial section of endometrium we identified protein expression of genes upregulated in the fibroblast minor and fibroblast major clusters. **a** In endometrial tissue of women with and without endometriosis, we used cytokeratin (green) to identify and exclude epithelial cells. MMP3 (red) in the marker for the fibroblast minor cluster was expressed in epithelial cells and a select number of non-epithelial cells. **b** Inset shows higher magnification. Expression appeared more common in endometriosis patients compared to controls. The increased expression also appeared in the secretory stage of both endometriosis patients and controls compared to the proliferative stage. **c** Endometrial tissue of women with and without endometriosis also expressed ACTA2 (green), a marker for the fibroblast major cluster. Samples were also incubated with cytokeratin (red) to identify epithelial cells. Expression was detected in both epithelial (red) and non-epithelial cells. **d** Expression of ACTA was widespread across non-epithelial cells in both women with and without endometriosis and appeared stronger in samples derived from the proliferative stage.

analysis. Our immunofluorescent analysis of endometrial samples from women with and without endometriosis supported a focal expression of MMP3 with a more pronounced expression in the secretory stage and in women with endometriosis. Importantly this focal expression did not appear to overlap with the expression of ACTA2 a marker for the fibroblast major cluster. Studies in the skin have suggested *MMP10* expression is predominantly limited to epithelial cells[34], but has also been reported in endometriosis[35] and shown to control the immune response in macrophages[36]. MMP10 expression is increased in the bladder[37], oesophagus[38] and skin cancer[39] and has been shown to be instrumental in bladder tumour cell migration and invasion[40], and wound healing and matrix remodelling in skin cancer[39].

The fibroblast minor cluster was also characterised by a strong expression of *CST1*. *CST1* has limited expression in most tissues of the body[41,42], although deep proteome and transcriptome sequencing confirmed endometrial expression[43]. Upregulation of *CST1* has been observed in malignant tumours and is associated with cancer cell proliferation, invasion and tumour recurrence[44–46]. The combination of extracellular matrix with upregulation of these genes may provide the fibroblast minor cluster, through its enhanced adhesion and infiltration capabilities, the capacity to establish lesions and thus increase disease susceptibility.

*CST1* has also been proposed as a fibroblast senescence marker[47]. Cellular senescence is a state of permeant cell-cycle arrest and is accompanied by the secretion of extracellular matrix proteins, proinflammatory cytokines and growth factors[48]. The presence of senescent decidual endometrial stromal cells has been

observed both in vitro and in vivo[49]. Decidual endometrial stromal fibroblasts differentiate from stromal fibroblasts approximately mid cycle and in response to rising progesterone concentrations. It was recently shown that an acute stress response precedes decidualisation leading to either decidualised, or senescent decidual cells[50]. In our study, we show altered differentiation of mesenchymal cells initiated prior to exposure to an acute stress response resulted in a subset of fibroblasts with high expression of senescent marker CST1. It is possible an altered gene expression profile developed prior to decidualisation could predispose some stromal fibroblasts to follow a divergent pathway during decidualisation resulting in cells with an altered gene expression profile and functional activity. Senescent DCs could be one such divergent outcome. Single-cell sequencing of cultured endometrial stromal cells identified the emergence of this subset of senescent decidualised stromal cells and found they were linked to aberrant endometrial biology, increasing susceptibility to recurrent pregnancy loss[50].

The fibroblast minor cluster we observe with altered immune reactivity may have a corollary to these subsets of decidualised endometrial cells produced from divergent differentiation pathways. Variation in the transcriptome of decidualised cells was observed that were acquired during maturation and were dependent on gene expression profiles of the starting cell. Isolated SUSD2+ and SUSD2- endometrial stromal cells that underwent differentiation to decidualising stromal cells retained distinct transcriptomic profiles that were characterised by differences in the secretion of inflammatory mediators with the decidualised SUSD2+ cells producing significantly more leukaemia inhibitory

factor (LIF) and chemokine ligand 7 (CCL7) compared to the decidualised SUSD2- stromal fibroblast[4].

Information from this study may also contribute to the understanding of endometriosis progression and not just pathogenesis. Recent research is beginning to identify the importance of fibrosis in endometriotic lesions, influencing both disease progression and treatment (51,52). Gli1+ marks perivascular MSC-like cells that contribute to organ fibrosis and in endometriosis, the immune environment of the peritoneal cavity can stimulate fibrosis through smooth muscle metaplasia (SMM) of endometrial stromal cells (51,52). The identification of the smooth muscle cells in this dataset supports this hypothesis and may represent a further progression of the differentiation pathway that can be induced when exposed to variations in the extracellular environment.

The identification of an aberrant differentiation pathway of a key cell type within the endometrium raises the exciting potential to understand disease susceptibility and generate diagnostic and prognostic biomarkers and generate novel targets for treatment. It should be cautioned however that this remains an in vitro study that may limit its potential translation into clinical effectiveness. Primarily, a potential confounding factor is the removal of these cells from their in vivo environmental niche, which affects the external influences that may control cellular function, development and differentiation. Whether these same subsets of endometrial stromal fibroblasts cells are still generated in an in vivo situation cannot be answered in this study and should be addressed in the future. The fact that this mechanism has been identified in vitro, and removed from the normal environment, does not invalidate the potential for these cells to harbour the potential to take a divergent differentiation pathway. In fact, it could be argued a similar process, after the removal from the niche eutopic environment during menstruation occur once in the ectopic environment and may initiate cues for this aberrant differentiation programming to begin.

In addition, in vitro growth assays indicated differences in the growth properties of each cell preparation, a characteristic commonly identified in primary cell cultures, and a limitation in identifying consistent functional characteristics using cell preparations from different patients. By charting the correlation of cell types identified in our in vitro study we could identify distinct growth profiles based on the variations in the proportion of different cell types that were initially isolated. These growth profiles appear to endure even after long periods in culture, supporting the view that important divergence in cellular differentiation can lead to long term changes in function. Whether the relative proportion of each of these cell types remains constant as the cells age, or whether increasing passage manifests in alternate differentiation trajectories is not assessed in this study but deserves further attention. While it is not clear whether the gene expression signatures identified directly influence the function of these individual cell preparations or relate to endometrial pathologies, the gene expression signatures generated from the single-cell sequencing strongly indicate variations in cell content even in FACS cell populations that correlate with functional differences.

In addition, as an in vitro study, other technical effects could also contribute. These analyses were performed with cryopreserved cells that may alter gene expression. A recent systematic analysis of the influence of cryopreservation on single-cell gene expression found a significant variable influence on cell integrity based on the method of cryopreservation used. However the use of cryopreservation media showed no discernible effect on cell integrity and impurities, measures of scRNA-sequencing transcript quality, and gene expression profiles of $R = 0.98$, with no influence on storage time[51]. It is also possible cryopreservation may bias cell composition through an increased sensitivity to the

freeze-thaw process for some rare cell subsets, which should be considered. In our study, there was no variation in methods applied based on clinical parameters and is thus unlikely to bias the association with endometriosis.

In addition, validation studies will be vital in confirming the existence of these cells in women with endometriosis. Appropriate validation approaches will however need to be carefully selected, designed and optimised. The fibroblast cluster associated with endometriosis is based on a combination gene signature, with large numbers of genes, in highly similar fibroblasts cells. A challenge will be to identify a single, or a couple of markers that may be used for diagnostic purposes. Furthermore, these signatures are based on subtle differences in expression that may be difficult to quantify clinically. Whether these genes are actively translated into proteins at detectable levels will also be relevant for the development of cost-effective diagnostic tools. Most importantly, the data from the study indicates these signatures arise from aberrant differentiation. Understanding when to look for and examine this aberrant differentiation will be vital for effective translation. Irrespective of these limitations and future work required this study raises important questions about the dynamic nature of each endometrial cycle and the potential that risks change each cycle.

In summary, previous work on endometriosis has suggested significant differences in the endometrium of women with and without endometriosis, although the mechanisms behind these variations and their contribution to endometrial pathologies is yet to be fully elucidated. Cell heterogeneity derived from variations in cell states or altered maturation pathways is common and may be embedded during cell fate lineage determination and can be leveraged by disease processes. By analysing endometrial stromal cells at a single-cell level with sufficient cell numbers, depth of sequencing and appropriate resolution we have uncovered a divergent mesenchymal differentiation of stromal fibroblasts that is significantly more likely to occur in cells from women with endometriosis. This could increase the susceptibility of the cells of some women to initiate endometriosis lesions at any particular cycle and may represent a potential biomarker. Divergent differentiation of stromal fibroblasts may provide novel targets for future treatment paradigms and warrants further investigation.

## Methods

**Sample collection.** Prior to surgery the relevant institutional review board granted ethical approval for the collection of samples and informed consent was obtained from all patients. Exclusion criteria for the study included abnormal ovulatory menstrual cycles and the use of either hormonal medication in the past 3 months. Patients with prior or current infections and liver dysfunction were also excluded. Prior to surgery whole blood was extracted and serum extracted. For the endometrial stromal cell preparations, the menstrual stage was assessed via the measurement of progesterone concentrations using a standard immunoassay. Patients were assigned to 1 of 3 stages based on the serum progesterone concentration; proliferative (0.181–2.84 nM), periovulatory (2.84–5.82 nM), or secretory (>5.82 nM). Prior to laparoscopy endometrial biopsies were collected via soft curette (Pipelle de Cornier, Laboratorie CCD, France) and stored in Complete IMDM media (10% fetal calf serum (FCS), 1% antibiotics/antimycotics (Invitrogen Life Technologies)) supplemented with 10% dimethyl sulfoxide (DMSO) (Thermo Fischer Scientific, Waltham, MA, USA) using the slow freezing method in a Bicell vessel at −80 °C. For analysis of protein expression in endometrial samples via immunofluorescence endometrial curettes were collected and samples sent to pathology for histopathological analysis and menstrual cycle dating and the remaining tissue stored flash frozen in liquid nitrogen until further use. The pelvic cavity of each patient was subsequently examined, any endometriotic lesions removed and the patient staged according to the revised American Fertility Society staging system (rAFS)[52].

**Endometrial stromal cell preparation.** The endometrial stromal cells were prepared as described previously[53]. Briefly, the tissue was thawed at 37 °C, washed with serum-free medium to remove DMSO and dissected into smaller pieces. The tissue was washed in phosphate-buffered saline (PBS) and incubated for 90 min at 37 °C in the presence of collagenase (10 mg/ml, Sigma) and subsequently filtered through 100 μm mesh (Falcon) to remove debris and undigested material. This was

followed by a second filtration through 40 μm mesh which will retain intact epithelial glands and allow individual stromal cells to pass through. Two volumes of IMDM was immediately added to the filtrate containing single stromal cells. The cells were centrifuged $5 \times 500\,g$, the supernatant discarded and the pellet resuspended in 1 ml fresh complete IMDM. Cells were seeded into 75 cm² flask for propagation.

Cells were maintained in culture using complete media (IMDM, 10% FCS, 1% antibiotic/antimycotic). Growth curves and cell viability were monitored via the recording of population doubling and cells maintained in a proliferative state by passaging using a standardised 1:3 split with trypsin/EDTA when cells were ~80% confluent. Once sufficient cell stocks were grown for subsequent experiments cells were trypsinized and counted using the automated Countess Cell Counters (Thermo Fisher Scientific). For cell freezing, cells were trypsinized, collected and centrifuged for 5 min at $1000\,g$. Cells pellets were resuspended in chilled (4 °C) cryopreservation media (Complete IMDM media with 10% DMSO (v/v) at $4 \times 10^6$ cells per vial. Vials were placed in a CoolCell (Merck) slow freezing container and stored at −80 °C overnight, after which the cells were transferred to liquid nitrogen storage.

**Cell thawing, FACS selection and sample pooling**. All samples were removed from liquid nitrogen, thawed and washed twice in IMDM complete media. Directly after thawing, cells were assessed for viability and their concentration was determined by manual cell counting. Cells were diluted in PBS to a final concentration of $2 \times 10^6$ cells into 50 μl of PBS and were taken directly for FACS sorting for viable and PDGRFβ+ cells. Positive PDGRFβ was used for sorting to ensure the analysis was restricted to cells of mesenchymal lineage, as it is a marker for mesenchymal cells present early during maturation and continues to be expressed throughout development into mature mesenchymal derived stromal cells[54].

For sorting the cells were incubated in blocking buffer (PBS, 40% FCS, 1% BSA) for 30 min and subsequently incubated with the mouse monoclonal anti-human PDGRFβ+ antibody conjugated to BV786-A (Becton Dickinson Cat No; 743038) with a 1:37.5 dilution in PBS, 10% FCS and 1% BSA for 1 h. Prior to cell sorting, 2 μl of propidium iodide (PI) was added to each sample. FACS was performed using the Aria II FACS machine (Becton Dickinson) with a dual-colour setting to select PDGFRβ + positive cells via dedicated excitation and emission settings for Brilliant Violet 421 and cells that excluded PI. Sorted cells were collected in PBS containing 10% FCS, and were then counted and their viability determined by haemocytometer and Trypan Blue staining. Cells with viability <80% were excluded from further analysis.

Pools of cells from multiple patient samples were generated prior to loading the 10x Genomics Chromium microfluidic chip channels. To obtain a final concentration of 20,000 cells per pool with an equimolar concentration of cells from each sample, we aimed to pool samples as follows: for pools of five samples, we added ~8000 cells from each sample, and for pools of four samples we added ~10,000 cells per sample. This yielded a final count of ~20,000 cells per pool due to the expected loss of cells during microfluidic processing.

**Genotyping and imputation**. DNA samples were isolated from the cell cultures and were genotyped using the Infinium Global Screening Array (Illumina Inc, San Diego). Quality control of genotypes was performed using PLINK[55] and SNPs with a missing rate of >5%, minor allele frequency (MAF) $< 1 \times 10^{-4}$ and with a Hardy-Weinberg Equilibrium (HWE) $p < 1 \times 10^{-6}$ were removed, leaving 645,726 SNPs for imputation. Imputation was performed using the 1000 Genomes Phase 3 reference panel. Genotyping data were used to identify the ancestry of each patient using 1000 Genome genotype data and principal component analysis.

**Single-cell RNA-sequencing and analysis**. The FACS single-cell suspensions were used to generate barcoded single-cell 3′ cDNA libraries for each of the pools with the Chromium Single-cell 3′ Gel Bead and library kit v2 (10x Genomics). Library quality control was performed with the Agilent Bioanalyzer High sensitivity DNA chip (Agilent). Denatured libraries were loaded onto an Illumina Nova-Seq6000 and sequenced with a $2 \times 100$ base-pair output for an average depth of 54,321 reads/cell.

The cellranger pipeline (v3.0.2) was used to process the sequencing data that included the mkfastq, count and aggr functions. The raw Illumina base call files were demultiplexed into sample-specific FASTQ files using cellranger mkfastq. Quality control (QC) was performed on the sample-specific files and subsequently aligned to the hg38 human reference using STAR[56] within the cellranger count algorithm. Aligned reads were filtered for valid cell barcodes and unique molecular identifiers, and resulting count matrices were combined into a single dataset using the cellranger aggr function. SNP genotyping data were used to identify doublets, multiplets and ambient cells using the Demuxlet software[26]. The remaining cells were taken forward for further analysis using the Seurat package (v3.0.2) in R (v3.4.1). We applied the following QC and filtering steps to the raw data: exclude (i) cells with >10% mitochondrial gene expression, (ii) cells with very low (<200) or very high (>6,500) numbers of expressed genes and (iii) genes expressed in very small numbers of cells (≤3). Between-cell gene expression was normalised using scTransform[57]. Between-pool variation due to technical and biological differences was corrected using filtered and normalised data with Harmony[58].

**Bioinformatic analysis**. Seurat was subsequently used to perform Louvain clustering of cells with the first 50 principal components and using a parameter sweep across multiple resolutions between 0.01 and 1.0. Cluster stability was assessed using clustree[16] and, based on clustering stability, clustering information from resolution 0.1 was retained for analysis. Differentially expressed genes (DEGs) between each cluster were determined with the Wilcoxon rank-sum test in Seurat with a minimum per cent expressing cells ≥0.25 and minimum absolute log fold-change threshold ≥ 0.25. Gene expression differences were considered significant if the adjusted $p$-value was $<1 \times 10^{-4}$ (Benjamini-Hochberg correction for multiple testing) and the absolute log-fold expression changes ≥0.5. Pathway enrichment analysis was performed with the top 200 DEGs in each cluster using the EnrichR package[59]. The enrichment ranking for pathways, ontologies, transcription factor networks and protein network analysis was calculated from the multiplication of a log p-value from the Fisher exact test by the Z-score of the deviation of the expected rank.

To identify the potential cell types within the dataset, a transcriptome-based cell-type classification was performed with SingleR[20] interrogating the Human Primary Cell atlas (HPCA) and the Blueprint+Encode reference datasets. Cell fate trajectory was predicted using the Monocle 2 package's pseudotime analysis[22] using the 500 genes with the highest variation in expression across all cells. For the Monocle 2 analysis, variation in gene expression was determined using the FindVariableFeatures function in Seurat and genes were ranked from the most to least variable. The cellular trajectory was further analysed via RNA velocity using dynamic modelling with velocyto[24] and scVelo[60].

**Demultiplexing patient samples from single-cell pools**. Sample demultiplexing was performed using the Demuxlet software[26] to assign cells to genotyped individuals and identify doublets. The position-sorted BAM file produced by the cellranger count function and a VCF file containing the genotype information for each sample were used as input into Demuxlet, where each cell barcode was assigned to a specific sample (or a pair of samples) in the VCF file using the genetic variation sequenced in each cell.

**Real-time analysis of cell adhesion and proliferation**. Additional cells that were not processed for RNA sequencing were maintained in culture for additional passages of up to 20 passages. Cell preparations that remained in a log phase of growth, as determined by population doubling calculations, were used to assess growth characteristics between passages 12–14. Cells were trypsinized and counted using the methods described above. Sixteen-well E-plates (ACEA Bioscience) were pre-incubated with 50 μl of prewarmed media and allowed to equilibrate in the incubator at 37 °C with 5% $CO_2$ for 60 min. Each well per plate was inoculated with 10,000 cells in a total volume of 100 μl, as this was previously determined as the optimal seeding density for cellular proliferation. The xCELLigence RTCA was set to perform a complete sweep across the plate to record cell growth, as Cell Index, every 15 min. Growth was profiled over a 100 hour period. Cell index data were normalised at the first time point post seeding to account for variations in cell concentrations, and exported for statistical analysis in Graphpad Prism v8.

To assess whether the relative proportions of cells identified in the original primary cell preparation could be associated with functional characteristics we calculated two values. Firstly, we determine the percentage of cells in each in vitro preparation that were assigned to a cell type, or cluster. This represented the relative proportion of each cell type in each preparation. Secondly, to quantify changes in functional characteristics we identified the number of cells of each preparation present at continuous 15 min intervals after being freshly seeded onto a real time xCELLigence growth plate. Finally, to determine the association between these two variables (per cent of cell content and number of cells) at each time point we calculated their correlation (Pearson's $r$) and plotted this correlation value ($r2$) against each time point.

**Immunofluorescence on endometrial samples**. Endometrial curettes were collected from 20 patients. This included 10 women with endometriosis, 5 collected during the secretory stage and 5 collected during the proliferative stage of the menstrual cycle. An additional 10 samples were collected from women without endometriosis, 5 of which were collected during the secretory stage and 5 of which were collected from the proliferative stage of the menstrual cycle. Frozen samples were embedded into Tissue Tek OCT (ProSciTech) and serial sections of 7 μm thickness were cut for each sample. Dual immunofluorescences were performed by incubation of sections with primary antibodies for either mouse anti-cytokeratin (Novus Biotechnie, 1:100) and rabbit anti-MMP3 (Abcam, 1:100), or rabbit anti-cytokeratin (Abcam, 1:100) and mouse anti-ACTA2 (Thermo Fisher Scientific, 1:100) for 1 h at 4 C overnight in a humidified chamber. The next day primary antibodies were removed, sections were washed and incubated with secondary antibodies Goat anti-Rabbit IgG (Alexa Fluor 647) (Abcam, 1:500) and Donkey anti-mouse IgG (Alexa Flour 488, 1:500) (Thermo Fisher Scientific) for both combinations of primary antibodies. Incubation was performed for 1 hour at room temperature after which sections were washed with Tris-buffered saline containing 0.1% Triton x-100 and mounted with DAPI-containing mounting media (ProSciTech) and a coverslip attached. Whole sections were digitised via scanning on the AxioScan Z1 Fluorescent Imager (Zeiss).

**Statistics and reproducibility**. All data have been presented unless otherwise stated as mean ± standard error of mean (SEM). Samples sizes for each experiment are indicated in the relevant results section or figure legends and represent biological replicates. Statistical analysis was performed in R (v3.4.1) and GraphPad Prism v8 software.

**Ethics approval and consent to participate**. Tissue sample collection was approved by the Cantonal ethics commission Bern (149/03) and the Metro North Human Research Ethics committee (2019/QRBW/56763). Experimental procedures were approved by the Cantonal ethics commission Bern (2019-01146) and the University of Queensland Human Research ethics committee (2016001723) (2019/HE002744).

**Reporting summary**. Further information on research design is available in the Nature Research Reporting Summary linked to this article.

## Data availability

In accordance with our HREC approvals at The University of Queensland and Bern Canton Ethics Committee and the open source policy for our funding for this study, only non-identifiable datasets are to be made public. Expression data (raw and normalized) and associated metadata were deposited in Zenodo and are available at https://zenodo.org/record/6572045. Data may also be made available upon reasonable request to the corresponding author. Differential expression gene lists are supplied in Supplementary Data files 1–6. Numerical source data used to generate graphical figures is available at Figshare[61].

## Code availability

All codes used for this manuscript are publicly available and have been cited in the text.

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

## Acknowledgements
We thank the staff of the Genome Innovation Hub, the University of Queensland for constructive discussions on the deconvolution of patient samples and the use of the Demuxlet software. This study was supported by a grant from the National Health and Medical Research Council (NHMRC) project grant GNT1147846. GWM was supported by NHMRC Fellowships GNT1078399 and GNT1177194.

## Author contributions
B.M. conceived of the project, performed experiments, analysed the data and prepared the manuscript. S.L. performed bioinformatic analysis and assisted with manuscript preparation. S.M. assisted with bioinformatic analysis. J.C., S.A., S.S., R.J. and K.T. assisted with experiments. K.N. and K.T. assisted with data and sample collection. A.A., M.D.M. and G.W.M. provided access to samples, equipment and funding.

## Competing interests
The authors declare no competing interests.
