## [Peer Review File · Communications Biology]

Reviewers' comments:

Reviewer #1 (Remarks to the Author):

McKinnon et al performed single cell sequencing on cells isolated from endometrial biopsies of 10 women with endometriosis and 9 women without endometriosis. A total of 33,758 cells were analyzed and clustering analysis revealed three clusters. Analysis was performed to identify the cell types, consisting mostly of fibroblast and mesenchymal stem cells (MSC). Cluster 0 and 2 consisted primarily of fibroblast and cluster 1 consisted mostly of MSC. Pseudotime and RNA velocity analysis show a progression from the MSC population towards cluster 1. The cells from cluster 2 appeared to have a slightly different, but parallel differentiation flow in the RNA velocity analysis, while the pseudotime analysis had ambiguous results for cluster 2 cells. The fibroblast split into two different populations, those from cluster 0 and those from cluster 1. The fibroblasts in cluster 2 were enriched in women with endometriosis, which suggests that fibroblasts with an aberrant differentiation program, are enriched in women with endometriosis. Proliferation assays were performed on the patient cells and suggest that there are differences in the growth patterns of the fibroblasts from cluster 0 and those from cluster 1.

Major points:

- 1) There idea that there is a unique population of fibroblasts with an altered differentiation state in women with endometriosis is very exciting and impactful. However, this needs to be validated by another method. Some of the genes found to be enriched in cluster 2 (Fig 2A) should be validated using IHC in patient samples to determine whether this population can be observed.
- 2) The color scheme to identify the various clusters appears to change between figures, making it difficult to follow. This is particularly confusing in the RNA velocity plot. But also the clusters are colored differently between Fig 4A and 4E.
- 3) Figure 5 was confusing and the results seem to be over-reaching. The experimental set up and the way the analysis was done should be explained better. In particular, at which point the cell population was assessed (was this taken from the single cell data and therefore is just the starting population?) and how the correlation was calculated. As well, limitations to this approach should be noted.

Minor points:

- 1) Culturing the patient cells will inevitably lead to some in vitro selection. This should be acknowledged in the text.
- 2) It would be interesting to show the age of the patients within the two groups, as age may be a confounding factor.

Reviewer #2 (Remarks to the Author):

McKinnon et al present a study using scRNASeq to analyse the mesenchymal compartment of the endometrium in women with and without endometriosis. The authors show 3 populations of mesenchymal cells; MSCs, and 2 fibroblast clusters. One population of fibroblasts seems exhibits incomplete differentiation, an altered growth profile and predominately evident in the cultures derived from women with endometriosis.

It is an interesting study with thoughtful bioinformatic analysis and a good amount of samples. My main criticism is that it is written in a conservative manner with very little interpretation of the findings and the authors may want to consider working on selling their main findings such that they shine through in the manuscript more predominantly..

Main points:

1. Please can the authors include more information on the cells in the methods. How were the cells frozen, passage number, media details, were they analysed straight from thawing? etc. These details are all important for reproducibility.
2. Patient characteristics are missing.... especially cycle phase. Is this a mix of proliferative and secretory phases? Were they evenly distributed across cases and controls?
3. Which samples were pooled into which group? Mix of cases and controls in each?
4. The authors show differences in clustering resolution. They show two extreme course and fine, what if the resolution is increased only slightly? Can they ever separate a MSC only cluster?
6. Fig.5 the difference in cell growth... is this comparative to what would be seen with a clonogenicity assay?
7. Results sections... can these headings be short conclusions? Also found the results section was a bit cryptic in places, can a 1 sentence summary of main finding and a little interpretation be added in each section to aid the narrative?
8. Conclusion: lines 362-370 about senescence, how exactly does this relate to you findings, particularly if you are suggesting these cells are less differentiated. Can you extrapolate your findings more in relation to this? I was a little confused by this.. Did you look at expression of Dio2 or Scara5? How do the concurrent findings of increased senescence with decreased endometrial plasticity fit with your findings of?
9. Can the authors clarify the findings that there is +ve correlation between msc content and cell index (line 280) BUT the fibroblast minor cluster (which has more mscs in it) has a negative correlation.
10. Line 374-380. Do you mean previous findings?

Reviewer #3 (Remarks to the Author):

The present paper "Altered differentiation of endometrial mesenchymal stromal fibroblast is associated with endometriosis susceptibility" aims to understand the link between endometriosis and the incomplete differentiation of the fibroblast, as part of disease susceptibility. The purpose of the authors constitutes an interesting approach. However, the weakness of this research lies on the study design and the management of the data obtained, certain technical aspects of the paper raise a number of questions about the validity and consistency of the results that should be address by the authors.

1. The authors decided culture the cells, cryopreserve and thawing them and after that perform the single cell analysis, Why? . This decision imposes an important bias, because there is no way to dissect out the transcriptomic difference impose by the disease, the cryopreservation/thawing strategy or the cell culture. For that these reason, this study is now an in vitro study and this concept has to be explained throughout the manuscript.
2. Why the authors take the decision to perform pools of samples. Using single cell approach is not necessary to pool the samples. It generates the lack of intraindividual variability essential to understand the question of the study. Please explain.
3. Could the authors explain why the trajectory inference from the 2 methods applied (Monocle2 and RNAvelocity) does not show similar results?.
4. How cluster package analysis for cluster stability analysis is performed over the dataset? is not explained.
5. Which is the marker profile of mesenchymal stem cells that the authors use?.
6. Enriched selection of PDGFRB cells is not properly explained . Why PDGFRB+ cells?.

Editor's Comments

We ask that you provide additional validation with IHC in patient samples following Reviewer 1's recommendation;

Response: We have conducted additional validation with immunofluorescence in patient samples for a marker of the fibroblast minor cluster (MMP3) and a marker of the fibroblast major cluster (ACTA2). The immunofluorescent analysis of endometrial samples from women with and without endometriosis addressed the request from Reviewer 1 with more pronounced expression of MMP3 in endometriosis patients especially in the secretory phase of the menstrual cycle (see our response to Reviewer 1). This focal expression did not appear to overlap with expression of ACTA2 a marker for the fibroblast major cluster. The effect of individual markers is expected to be relatively small because the cluster separation in the paper is a cell state defined by a gene signature from combined results of multiple genes (cluster 0 = 698 genes, cluster 1 = 963 genes and cluster 2 = 203 genes). Nevertheless, the immunofluorescence results for the *MMP3* and *ACTA2* support the result from the gene expression analyses.

Address Reviewer 3's concerns regarding data pooling and the effects of cryopreservation, whether through a discussion of its limitations on the conclusions or more preferably, by testing the gene expression of a non-cryopreserved sample.

Response: As requested by the editor we have addressed the concerns of Reviewer 3 through additional discussion. Please see our response to specific comments from Reviewer 3 for further details.

Reviewers' comments:

We thank the reviewers for their comments and have provided a detailed response to each of the reviewers' comments below, including the text within the revised manuscript that is associated with the comment and the line at which it can be found. Please note the line number refers to the clean copy manuscript.

Reviewer #1 (Remarks to the Author):

McKinnon et al performed single cell sequencing on cells isolated from endometrial biopsies of 10 women with endometriosis and 9 women without endometriosis. A total of 33,758 cells were analyzed and clustering analysis revealed three clusters. Analysis was performed to identify the cell types, consisting mostly of fibroblast and mesenchymal stem cells (MSC). Cluster 0 and 2 consisted primarily of fibroblast and cluster 1 consisted mostly of MSC. Pseudotime and RNA velocity analysis show a progression from the MSC population towards cluster 1. The cells from cluster 2 appeared to have a slightly different, but parallel differentiation flow in the RNA velocity analysis, while the pseudotime analysis had ambiguous results for cluster 2 cells. The fibroblast split into two different populations, those from cluster 0 and those from cluster 1. The fibroblasts in cluster 2 were enriched in women with endometriosis, which suggests that fibroblasts with an aberrant differentiation program, are enriched in women with endometriosis. Proliferation assays were performed on the patient cells and suggest that there are differences in the growth patterns of the fibroblasts from cluster 0 and those from cluster 1.

Major points:

1) There idea that there is a unique population of fibroblasts with an altered differentiation state in women with endometriosis is very exciting and impactful. However, this needs to be validated by another method. Some of the genes found to be enriched in cluster 2 (Fig 2A) should be validated using IHC in patient samples to determine whether this population can be observed.

Response: We have performed immunofluorescence on markers from the fibroblast minor cluster (MMP3) and the fibroblast major cluster (ACTA2), as well as performing co-staining for cytokeratin to enable exclusion of endometrial epithelial cells from cells that show positive MMP3 or ACTA2 expression. These immunofluorescence results were consistent with our gene expression results, showing a more prominent expression of MMP3 in endometriosis samples and samples from the secretory stage, and importantly that MMP3 and ACTA2 were expressed by distinct sets of cells. This also supports previously published literature (1, 2).

We have updated the text of the results section and included new figures (Figure 6, Supplementary Figure 4 and Supplementary Figure 5) to report these additional findings.

Line 360-389: “MMP3 and ACTA2 are expressed by distinct sets of stromal cells in the endometrium of women with and without endometriosis

To assess whether the gene expression markers of the fibroblast minor (MMP3) and fibroblast major (ACTA2) clusters could be detected at the protein level and whether these makers were expressed in distinct cells of mesenchymal origin in the endometrium, we probed endometrial samples from women with and without endometriosis, isolated at either the proliferative or secretory stage of the menstrual cycle with specific antibodies for either MMP3, or ACTA2. Immunofluorescent images indicated only minimal MMP3 (red) expression in all endometrial samples (Supplementary Figure 4). To exclude epithelial cell expression of these proteins we also performed co-staining with cytokeratin (green) a marker specific for epithelial cells. (Supplementary Figure 4). Co-localisation with cytokeratin confirmed MMP3 expression in epithelial cells, however some non-epithelial, stromal cells were also positive for MMP3. MMP3 expression appeared strongest in both endometriosis and control samples during the secretory stage with a slightly more prevalent expression in the endometriosis compared with control samples (Figure 6A-B).

Incubation with serial sections of the same endometrial samples with ACTA2 (green) and cytokeratin (red) on adjacent slides (Supplementary Figure 5) also confirmed expression of ACTA2 in endometrial cells. Co-staining with cytokeratin (red) indicated the majority of ACTA2-positive cells were non-epithelial stromal cells, although in some cases epithelial cells also showed positive expression. The positive ACTA2 stromal cells appeared to be more common in the samples isolated from the proliferative phase compared to samples isolated from the secretory stage, irrespective of endometriosis status (Figure 6C-D). No comprehensive quantification was performed to confirm these changes in expression. Importantly, using serial sections, we confirmed that both MMP3 and ACTA2 were expressed in endometrial stromal cells in the endometrium. Furthermore, we noted MMP3 expression appeared restricted to only a small number of endometrial stromal cells, consistent with the small cluster we observed at the gene level and that co-expression of MMP3 and ACTA2 in endometrial stromal cells was not detected when viewed under higher magnification (Figure

6B and 6D). This supports the existence of independent endometrial stromal fibroblasts subtypes in endometrial samples.”

We have also included a discussion of these results this in the discussion section.

Line 442-446: *“Our immunofluorescent analysis of endometrial samples from women with and without endometriosis supported a focal expression of MMP3 with a more pronounced expression in the secretory stage and in women with endometriosis. Importantly this focal expression did not appear to overlap with expression of ACTA2 a marker for the fibroblast major cluster.”*

2) The color scheme to identify the various clusters appears to change between figures, making it difficult to follow. This is particularly confusing in the RNA velocity plot. But also the clusters are colored differently between Fig 4A and 4E.

Response: We thank the reviewer for picking up these differences. We have adjusted the colours to keep consistent through the manuscript.

We have adjusted the colours in Figure 3F (the RNA velocity plot) so that it reflects the colours used to denote each cluster (Cluster 0 = red, cluster 1 = green, cluster 2 = blue) in other figures that use this clustering.

In Figure 4A the different colours represent cells that are derived from different patients that were multiplexed into individual runs. These colours represent individual patients and do not delineate clusters. Subsequently, in Figure 4E the colours have been used to identify cells that come from either cases or controls (not cluster). As such the different colours reflect different underlying characteristics of the cells and thus we believe it is appropriate to use different colours for these two figures.

3) Figure 5 was confusing and the results seem to be over-reaching. The experimental set up and the way the analysis was done should be explained better. In particular, at which point the cell population was assessed (was this taken from the single cell data and therefore is just the starting population?) and how the correlation was calculated. As well, limitations to this approach should be noted.

Response: As requested we have included additional description of the experimental set up. Details have been included in the Methods on when the cell populations were assessed:

Line 706-709: *“Additional cells that were not processed for RNA sequencing were maintained in culture for additional passages of up to 20 passages. Cell preparations that remained in a log phase of growth, as determined by population doubling calculations, were used to assess growth characteristics between passages 12-14.”*

We have included how the association was determined in the Methods.

Line 719-728: *“To assess whether the relative proportions of cells identified in the original primary cell preparation could be associated with functional characteristics we calculated two values. Firstly, we determine the percentage of cells in each in vitro preparation that were assigned to a cell type, or cluster. This represented the relative proportion of each cell*

type in each preparation. Secondly, to quantify changes in functional characteristics we identified the number of cells of each preparation present at continuous 2 minute intervals after being freshly seeded onto a real time xCELLigence growth plate. Finally, to determine the association between these two variables (percent of cell content and number of cells) at each time point we calculated their correlation (Pearson' r) and plotted this correlation value (r²) against each time point. “

We have also included a discussion on the limitations of these methods in the Discussion section:

Line 514-527: *“Additionally, in vitro growth assays indicated differences in the growth properties of each cell preparation, a characteristic commonly identified in primary cell cultures, and a limitation in identifying consistent functional characteristics using cell preparations from different patients. By charting the correlation of cell types identified in our in vitro study we could identify distinct growth profiles based on the variation in the proportion of different cell types that were initially isolated. These growth profiles appear to endure even after long periods in culture, supporting the view that important divergence in cellular differentiation can lead to long term changes in function. Whether the relative proportion of each of these cell types remain constant as the cells age, or whether increasing passage manifests in alternate differentiation trajectories is not assessed in this study but deserve further attention. While it is not clear whether the gene expression signatures identified directly influence the function of these individual cell preparations or relate to endometrial pathologies, the gene expression signatures generated from the single cell sequencing strongly indicates variations in cell content even in FACS cell populations that correlates with functional differences.”*

Minor points:

1) Culturing the patient cells will inevitably lead to some in vitro selection. This should be acknowledged in the text.

Response: We have added a discussion of the limitations of our *in vitro* studies to the Discussion:

Line 499–512: *“The identification of an aberrant differentiation pathway of a key cell type within the endometrium raises exciting potential to understand disease susceptibility, identify diagnostic and prognostic biomarkers and generate novel targets for treatment. It should be cautioned however that this remains an in vitro study that may limit its potential translation into clinical effectiveness. Primarily, a potential confounding factor is the removal of these cells from their in vivo environmental niche, which affects the external influences that may control cellular function, development and differentiation. Whether these same subset of endometrial stromal fibroblasts cells are still generated in an in vivo situation cannot be answered in this study and should be addressed in the future. The fact that this mechanism has been identified in vitro, removed from the normal environment, does not invalidate the potential for these cells to harbour the potential to take a divergent differentiation pathway. In fact, it could be argued a similar process, after the removal from the niche eutopic environment during menstruation occur once in the ectopic environment and may initiate cues for this aberrant differentiation programming to begin. “*

Further discussions on the limitations of our approach have been included in the Discussion:

Line 514-527: *“Additionally, in vitro growth assays indicated differences in the growth properties of each cell preparation, a characteristic commonly identified in primary cell cultures, and a limitation in identifying consistent functional characteristics using cell preparations from different patients. By charting the correlation of cell types identified in our in vitro study we could identify distinct growth profiles based on the variation in the proportion of different cell types that were initially isolated. These growth profiles appear to endure even after long periods in culture, supporting the view that important divergence in cellular differentiation can lead to long term changes in function. Whether the relative proportion of each of these cell types remain constant as the cells age, or whether increasing passage manifests in alternate differentiation trajectories is not assessed in this study but deserve further attention. While it is not clear whether the gene expression signatures identified directly influence the function of these individual cell preparations or relate to endometrial pathologies, the gene expression signatures generated from the single cell sequencing strongly indicates variations in cell content even in FACS cell populations that correlates with functional differences.”*

2) It would be interesting to show the age of the patients within the two groups, as age may be a confounding factor.

Response: We have included additional data on patient characteristics in the Results section titled: *“Pure, single cells of mesenchymal origin were isolated from women both with and without endometriosis.”* This includes data on the age of patients:

Line 120-127: *“Serum progesterone measurements were available for all 19 patients and age and BMI were available for 18 patients. Nine endometrial stromal cell preparations were isolated during the proliferative phase (7 x cases and 2 x controls), four were isolated during the periovulatory period (1 x case and 3 x controls) and six during the secretory stage (2 x cases and 4 x controls). No significant difference in menstrual cycle phases was observed between cases and controls (Supplementary Table 1). The average age and BMI of all patients was 35.17 ± 1.71 and 24.16 ± 1.06 respectively. There was no significant difference in age (case = 35.17 vs control = 35.0, $p = 0.939$), although there was a significant difference in BMI (case = 24.16 vs control = 27.29).”*

The data are also summarised in the **Supplementary Table S1**.

Reviewer #2 (Remarks to the Author):

McKinnon et al present a study using scRNASeq to analyse the mesenchymal compartment of the endometrium in women with and without endometriosis. The authors show 3 populations of mesenchymal cells; MSCs, and 2 fibroblast clusters. One population of fibroblasts seems exhibits incomplete differentiation, an altered growth profile and predominately evident in the cultures derived from women with endometriosis.

It is an interesting study with thoughtful bioinformatic analysis and a good amount of samples. My main criticism is that it is written in a conservative manner with very little

interpretation of the findings and the authors may want to consider working on selling their main findings such that they shine through in the manuscript more predominantly..

Response: We sincerely thank the reviewer for the enthusiastic opinion of our data. We believe that the data presented in this study is significant and may have important implications. We also believe, as discussed with reviewer 1 that additional confirmation of these findings will be important. We present the data as generated, plan additional work and remain cautiously optimistic about the implications of what we present within this manuscript.

Main points:

1. Please can the authors include more information on the cells in the methods. How were the cells frozen, passage number, media details, were they analysed straight from thawing? etc. These details are all important for reproducibility.

Response: Additional details on the cell preparations used for this study are provided in the Methods section, including information on freezing method; passage number, media details and thawing for analysis.

Line 607-612: *“For cell freezing, cells were trypsinized, collected and centrifuged for 5 mins at 1,000g. Cells pellets were resuspended in chilled (4 °C) cryopreservation media (Complete IMDM media with 10% DMSO (v/v) at 4×10^6 cells per vial. Vials were placed in a CoolCell (Merck) slow freezing container and stored at -80 °C overnight, after which the cells were transferred to liquid nitrogen storage.”*

Line 615-621: *“All samples were removed from liquid nitrogen, thawed and washed twice in IMDM complete media. Directly after thawing, cells were assessed for viability and their concentration determined by manual cell counting. Cell were diluted in PBS to a final concentration of 2×10^6 cells in 50 μ l of PBS and were taken directly for FACS for viable and PDGFR β + cells. Positive PDGFR β was used for sorting to ensure analysis was restricted to cell of mesenchymal lineage, as it is a marker for mesenchymal cells present early during maturation and continues to be expressed throughout development into mature mesenchymal derived stromal cells (53)”*

Additional passage details are summarised for each sample with other clinical characteristics presented in **Supplementary Table 2**.

2. Patient characteristics are missing.... especially cycle phase. Is this a mix of proliferative and secretory phases? Were they evenly distributed across cases and controls?

Response: Additional details have been provided in **Supplementary Table 1**. This includes additional data on the menstrual cycle stage at when cells were isolated, as well as age and BMI. These variables have also been assessed to determine if they were associated with gene expression, or cell numbers (**Supplementary Table 8** and **Supplementary Table 9**). No significant association was observed.

Line 290- 301: *“We wished to identify biological or clinical variables from our sample set that correlate with either cell type, or cell clustering. Splitting the cells based on the two major cell types identified, we assessed the correlation between gene expression and other*

variables including number of passages, menstrual stage at time of sample collection, patient age and endometriosis subtype. Almost all factors had a correlation close to zero and were well below the cut off value ($r^2 > 0.3$) to suggest any association with gene expression (Supplementary Table 8). We also assessed whether any of these characteristics could be associated with the cell numbers within each cluster. We found no significant association between the number of cells in each cluster with age (Cluster 0: $p = 0.3306$, Cluster 1: $p = 0.7670$, Cluster 2, $p = 0.8552$), or BMI (Cluster 0: $p = 0.4386$, Cluster 1: $p = 0.1352$, Cluster 2, $p = 0.9794$) nor was there an association between cell passage with any of the clusters identified (Cluster 0: $p = 0.1906$, Cluster 1: $p = 0.4724$, Cluster 2, $p = 0.1408$) (Supplementary Table 9).”

3. Which samples were pooled into which group? Mix of cases and controls in each?

Response: The case and control samples were mixed across each pool. The number of case and controls samples for each pool have now been included in the Results section.

Line 142-144: “Endometriosis cases and controls were distributed across each pool (P1: Control = 2, Case = 4, P2: Control = 3, Case = 2, P3: Control = 3, Case = 2, P4: Control = 2, Case = 2).”

4. The authors show differences in clustering resolution. They show two extreme course and fine, what if the resolution is increased only slightly? Can they ever separate a MSC only cluster?

Response: We thank the reviewer for the interesting question. We have performed some additional analysis on the clusters generated downstream but were unable to identify a cluster that is represented purely by MSCs.

6. Fig.5 the difference in cell growth... is this comparative to what would be seen with a clonogenicity assay?

Response: Thank you very much for the interesting question. We believe the real time data collected for this study contains inherent differences to the data that would be collected by performing clonogenic assays. By monitoring cells in real time, we can assess changes in growth rates as cell adapt to their environment. It records subtle changes in growth rates that may be transient. A clonogenic assay is more likely to identify the ability of cells to continue to proliferate medium to long term. We would be very interested in performing clonogenic assays on the different cell cluster if we can isolate these cells from the others cell clusters.

7. Results sections... can these headings be short conclusions? Also found the results section was a bit cryptic in places, can a 1 sentence summary of main finding and a little interpretation be added in each section to aid the narrative?

Response: As requested we have rearranged the heading to be short conclusions and added 1-2 sentence in each section to illustrate the main findings. This can be found at

Title: “Pure, single cells of mesenchymal origin were isolated from women with and without endometriosis”

Conclusion: *“Through this process we were able to prepare single endometrial cells of mesenchymal origin from 19 patients which were subsequently transcriptome-wide gene expression profiling.”*

Title: *“Single cell RNA-sequencing and assessment of cluster resolution identified three consistent subtypes of endometrial mesenchymal cells”*

Conclusion: *“Through the inclusion of multiple patients, multiplexing of samples and deep sequencing of expressed genes we were able to sequence a dataset of sufficient size and quality to identify three stable clusters of distinct mesenchymal cells.”*

Title: *“Cell cycle scoring indicates cluster 1 harboured an increased portion of proliferating cells”*

Conclusion: *“This analysis indicated most cells analysed were in the G1 phase and cell cycle was not directly associated with cell clusters.”*

Title: *“Differential gene expression between clusters reveals discrete signatures indicative of varied interaction with the microenvironment”*

Conclusion: *“This analysis indicated gene expression profiles of cluster 0 and 2 were associated with extracellular organisation, with cluster 0 focussed on adhesion, whereas cluster 2 may be influenced by immune response. Cluster 1 appeared specifically related to reproductive development.”*

Title: *“Cell-type annotation confirmed gene expression signatures consistent with cells of mesenchymal origin”*

Conclusion: *“Analysis of gene expression signatures therefore supported the mesenchymal lineage of the cell dataset, but also indicated subtle differences exist within cells that can be used to delineate variations”*

Title: *“Cell fate trajectory analysis identified the degree of mesenchymal differentiation for each cell”*

Conclusion: *“Comparison of the degree of gene expression change in each cell identified cluster 1 as the root cells extending to cluster 0 as the terminal differentiation. Cluster 1 existed across this trajectory and represented either incomplete, or dedifferentiated mesenchymal cells.”*

Title: *“Deconvolution of pooled samples successfully assigned each cell to the patient of origin”*

Conclusion: *“Using this method the patient of origin for each cell and the number of cells analysed for each patient in each pool was identified”*

Title: *“Comparison of gene expression profiles and cell clusters identified an association between the fibroblast minor cluster and endometriosis status in cultured stromal cells_”*

Conclusion: *“These data therefore indicate that the strongest association with gene expression profiles of the fibroblast minor cluster is the endometriosis status of the patient of origin.”*

Title: *“In vitro analysis confirmed cellular clusters displayed unique growth profiles”*

Conclusion: *“This analysis indicated that variations in the in vitro cell cultures identified by gene expression signature can still influence cell growth patterns even after placed in culture.”*

Title: “*MMP3 and ACTA2 are expressed by distinct sets of stromal cells in the endometrium*”

Conclusion: “*This supports the existence of independent endometrial stromal fibroblasts in endometrial samples.*”

8. Conclusion: lines 362-370 about senescence, how exactly does this relate to your findings, particularly if you are suggesting these cells are less differentiated. Can you extrapolate your findings more in relation to this? I was a little confused by this.. Did you look at expression of Dio2 or Scara5? How do the concurrent findings of increased senescence with decreased endometrial plasticity fit with your findings of?

Response: The previous work on cultured stromal cells revealed decidualisation is a multistep process that starts with an acute stress response. The cells undergo an intermediate transcriptional response and emerge predominantly as decidualised cells or senescent decidualised cells (3). It is possible an altered gene expression profile, developed prior to decidualisation could predispose some cells to follow a divergent pathway when exposed to the appropriate decidualisation stimulus, thereby increasing the likelihood they emerge with an altered gene expression and functional profile. One possible resulting profile could be senescence.

We have expanded the Discussion section to include this information:

Line 467-474: “It was recently shown that an acute stress response precedes decidualisation leading to either decidualised, or senescent decidual cells (50). In our study we show altered differentiation of mesenchymal cells initiated prior to exposure to an acute stress response resulted in a subsets of fibroblasts with high expression of senescent marker CST1. It is possible an altered gene expression profile developed prior to decidualisation could predispose some stromal fibroblast to follow a divergent pathway during decidualisation resulting in cells with a altered gene expression profile and functional activity. Senescent DCs could be one such divergent outcome.”

We have examined gene expression for DIO2 and SCARA5 and found DIO2 was significantly differentially expressed in Cluster 1 compared to all other cells and in particular to cluster 2. We found no differential expression of SCARA5 in the different clusters.

9.Can the authors clarify the findings that there is +ve correlation between msc content and cell index (line 280) BUT the fibroblast minor cluster (which has more mscs in it) has a negative correlation.

Response: In this comparison the fibroblast minor cluster contained 2.5% MSCs and the fibroblast major cluster had 0.83% MSCs. This is only a very small percentage of cells and not significantly different between the two clusters. We do not believe the amount of MSC in these clusters would be sufficient to influence the growth rates of all cells within each cluster.

10. Line 374-380.Do you mean previous findings?

Response: Thank you for the comment. Yes in this sentence we refer to previous findings. We have amended the sentence to show this meaning and also included some references.

Line 490-494: “Recent research is beginning to identify the importance of fibrosis in endometriotic lesions, influencing both disease progression and treatment (51,52). *Gli1+* marks perivascular MSC-like cells that contribute to organ fibrosis 53 and in endometriosis the immune environment of the peritoneal cavity can stimulate fibrosis through smooth muscle metaplasia (SMM) of endometrial stromal cells (51,52)”

Reviewer #3 (Remarks to the Author):

The present paper “Altered differentiation of endometrial mesenchymal stromal fibroblast is associated with endometriosis susceptibility” aims to understand the link between endometriosis and the incomplete differentiation of the fibroblast, as part of disease susceptibility. The purpose of the authors constitutes an interesting approach. However, the weakness of this research lies on the study design and the management of the data obtained, certain technical aspects of the paper raise a number of questions about the validity and consistency of the results that should be address by the authors.

1. The authors decided culture the cells, cryopreserve and thawing them and after that perform the single cell analysis, Why? . This decision imposes an important bias, because there is no way to dissect out the transcriptomic difference impose by the disease, the cryopreservation/thawing strategy or the cell culture. For that these reason, this study is now an in vitro study and this concept has to be explained throughout the manuscript.

Response: We thank you for your comment. We agree that there are differences between an *in vivo* and *in vitro* study. While certain aspects of *in vivo* environment will be lost, which is acknowledged throughout the manuscript, this does not exclude inherent variation remaining in cell function. We can point to studies showing a genetic predisposition toward endometriosis. These genetic variants will remain in cell culture and influence cellular function. Moreover, clear differences exist within cells taken from individual patients, whether differences due to individual pathologies can be observed can only be evaluated through study. The use of powerful single cell datasets allows us to assess whether differences occur. As requested, we have amended the main text in multiple locations to emphasis that this is an *in vitro* study and discussed these limitations. These limitations have been mentioned below:

Line 84-87: “To identify inherent variation in the endometrium that could underlie endometrial susceptibility we therefore assessed gene expression of individual mesenchymal-derived cells isolated and cultured from the endometrium and their association with clinical parameters and in vitro growth.”

Line 128-134: “Cells were frozen down between passages 4 – 7, with p4 = 4 (2 x case, 2 x control), p5 = 4 (0 x case, 4 x control), p6 (7 x case, 1 x control) and p7 = 1 (1 x case, 0 x control). A chi squared comparison indicated a significant difference between cases and controls ($p = 0.026$) with a slightly higher mean passage number for stromal cells from endometriosis cases, reflective of enhanced growth profile. Through this process we were able to prepare single endometrial cells of mesenchymal origin from 19 patients that were subsequently analysed transcriptome -wide gene expression profiling.”

Line 288-289: “Comparison of gene expression profiles and cell clusters identified an

association between the fibroblast minor cluster and endometriosis status in cultured stromal cells”

Line 499 – 512: “The identification of an aberrant differentiation pathway of a key cell type within the endometrium raises exciting potential to understand disease susceptibility and generate diagnostic and prognostic biomarkers and generate novel targets for treatment. It should be cautioned however that this remains an *in vitro* study that may limit its potential translation into clinical effectiveness. Primarily, a potential confounding factor is the removal of these cells from their *in vivo* environmental niche, which affects the external influences that may control cellular function, development and differentiation. Whether these same subset of endometrial stromal fibroblasts cells are still generated in an *in vivo* situation cannot be answered in this study and should be addressed in the future. The fact that this mechanism has been identified *in vitro*, removed from the normal environment, does not invalidate the potential for these cells to harbour the potential to take a divergent differentiation pathway. In fact, it could be argued a similar process, after the removal from the niche eutopic environment during menstruation occur once in the ectopic environment and may initiate cues for this aberrant differentiation programming to begin.”

Line 514-527: “Additionally, *in vitro* growth assays indicated differences in the growth properties of each cell preparation, a characteristic commonly identified in primary cell cultures, and a limitation in identifying consistent functional characteristics using cell preparations from different patients. By charting the correlation of cell types identified in our *in vitro* study we could identify distinct growth profiles based on the variations in the proportion of different cell types that were initially isolated. These growth profiles appear to endure even after long periods in culture, supporting the view that important divergence in cellular differentiation can lead to long term changes in function. Whether the relative proportion of each of these cell types remain constant as the cells age, or whether increasing passage manifests in alternate differentiation trajectories is not assessed in this study but deserve further attention. While it is not clear whether the gene expression signatures identified directly influence the function of these individual cell preparations or relate to endometrial pathologies, the gene expression signatures generated from the single cell sequencing strongly indicates variations in cell content even in FACS cell populations that correlates with functional differences.”

Line 529 – 538: “In addition, as an *in vitro* study other technical effects could also contribute. These analyses were performed with cryopreserved cells that may alter gene expression. A recent systematic analysis of the influence of cryopreservation on single cell gene expression found a significant variable influence on cell integrity based on the method of cryopreservation used. However the use of cryopreservation media showed no discernible effect on cell integrity and impurities, measures of scRNA-sequencing transcript quality, and gene expression profiles of $R = 0.98$, with no influence of storage time (54). It is also possible cryopreservation may bias cell composition through an increased sensitivity to the freeze-thaw process for some rare cell subsets, which should be considered. In our study there was no variation in methods applied based on clinical parameters and is thus unlikely to bias the association with endometriosis.”

2. Why the authors take the decision to perform pools of samples. Using single cell approach is not necessary to pool the samples. It generates the lack of intraindividual variability essential to understand the question of the study. Please explain.

Response: “We thank the reviewer for his comment, but would respectfully disagree. Pooling cells actually provides many benefits that significantly strengthens the study. Pooling allows the inclusion of multiple independent samples, as observed in the study. Sequencing cells from 19 different patients using a multiplexing approach allows the reduction in batch effects caused by running all samples in different wells, on different days and under variable conditions. Importantly this approach allows each cell to be traced to its original donor. Additionally, we have focussed this study on an individual cell type, rather than performing a wide cellular architecture study. The number of these cells, even with pooling still represents one of the larger studies on endometrial stromal cells to date. The large number of individual cell type studies should not be considered to influence intraindividual variability.”

3. Could the authors explain why the trajectory inference from the 2 methods applied (Monocle2 and RNAvelocity) does not show similar results?

Response: We have to respectfully disagree with this assessment. The trajectory analysis and the RNA velocity plot show very similar results. Both analyses show the movement of cells in cluster 1 differentiating towards cluster 0. Cluster 1 again in both analysis shows a directional movement consistent with a differentiation from cluster 1 to cluster 0 with an incomplete differentiation for this cluster.

4. How cluster package analysis for cluster stability analysis is performed over the dataset? is not explained.

Response: The cluster analysis stability is performed using the clustree software package. Additional information has been included in the Results section:

Line 164-170: “Using the clustree (16) package, we produced a cluster tree with 13 levels of resolutions ranging from 0.01-1.0 (Figure 1G) to visualise the similarity between cells at multiple resolutions, and track how cells move between clusters as resolution is varied. This package uses a hard clustering algorithms to cluster data at multiple resolutions producing a set of cluster nodes, the overlap between clusters is used to build edges and the resulting graph represents how each clusters relate to each other, which are distinct and which are unstable. This allows the visualisation and exploration of all possible choices (16)”

5. Which is the marker profile of mesenchymal stem cells that the authors use?.

Response: As discussed in the results section the mesenchymal stem cell profile was derived from the SingleR analysis of transcriptomic signatures available in the Human Primary cell atlas. This uses gene expression profiles and multiple genes based on previous studies that have contributed to the Human Cell Atlas. This represents a far more accurate identification of cell type than relying on individual cell markers that are historically generated using protein markers.

6. Enriched selection of PDGFRB cells is not properly explained . Why PDGFRB+ cells?.

Response: The purpose of our study was to focus on endometrial cells of mesenchymal lineage. As these are primary cells derived from endometrial tissue, we wished to ensure a pure population of mesenchymal cells, exclude any other cells that may be present (such as epithelial or immune cells) and to increase the power of our analysis using a selection marker

that would select for mesenchymal cells at their earliest origin. PDGFR β is a marker for mesenchymal cells that is expressed early in the mesenchymal cells and during maturation.

This has been included in the Methods section:

Line 619 -621: *“Positive PDGFR β was used for sorting to ensure analysis was restricted to cell of mesenchymal lineage, as it is a marker for mesenchymal cells present early during maturation and continues to be expressed throughout development into mature mesenchymal derived stromal cells (53).”*

References

1. Rodgers WH, Matrisian LM, Giudice LC, Dsupin B, Cannon P, Svitek C, et al. Patterns of matrix metalloproteinase expression in cycling endometrium imply differential functions and regulation by steroid hormones. *J Clin Invest.* 1994;94(3):946-53.
2. Rodgers WH, Osteen KG, Matrisian LM, Navre M, Giudice LC, Gorstein F. Expression and localization of matrilysin, a matrix metalloproteinase, in human endometrium during the reproductive cycle. *Am J Obstet Gynecol.* 1993;168(1 Pt 1):253-60.
3. Lucas ES, Vrljicak P, Muter J, Diniz-da-Costa MM, Brighton PJ, Kong C-S, et al. Recurrent pregnancy loss is associated with a pro-senescent decidual response during the peri-implantation window. *Communications Biology.* 2020;3(1):37.

REVIEWERS' COMMENTS:

Reviewer #2 (Remarks to the Author):

The authors have provided a thoughtful and comprehensive rebuttal, addressing all comments raised by the reviewers and including some additional data.

Reviewer #3 (Remarks to the Author):

The authors reply adequately every statement presented by the reviewer